# The Impact of Natural Dietary Compounds and Food-Borne Mycotoxins on DNA Methylation and Cancer

**DOI:** 10.3390/cells9092004

**Published:** 2020-08-31

**Authors:** Terisha Ghazi, Thilona Arumugam, Ashmika Foolchand, Anil A. Chuturgoon

**Affiliations:** Department of Medical Biochemistry, School of Laboratory Medicine and Medical Science, College of Health Sciences, University of KwaZulu-Natal, Durban 4041, South Africa; GhaziT@ukzn.ac.za (T.G.); 213531562@stu.ukzn.ac.za (T.A.); 215048184@stu.ukzn.ac.za (A.F.)

**Keywords:** cancer, epigenetics, DNA methylation, micronutrients, bioactive dietary compounds, mycotoxins

## Abstract

Cancer initiation and progression is an accumulation of genetic and epigenetic modifications. DNA methylation is a common epigenetic modification that regulates gene expression, and aberrant DNA methylation patterns are considered a hallmark of cancer. The human diet is a source of micronutrients, bioactive molecules, and mycotoxins that have the ability to alter DNA methylation patterns and are thus a contributing factor for both the prevention and onset of cancer. Micronutrients such as betaine, choline, folate, and methionine serve as cofactors or methyl donors for one-carbon metabolism and other DNA methylation reactions. Dietary bioactive compounds such as curcumin, epigallocatechin-3-gallate, genistein, quercetin, resveratrol, and sulforaphane reactivate essential tumor suppressor genes by reversing aberrant DNA methylation patterns, and therefore, they have shown potential against various cancers. In contrast, fungi-contaminated agricultural foods are a source of potent mycotoxins that induce carcinogenesis. In this review, we summarize the existing literature on dietary micronutrients, bioactive compounds, and food-borne mycotoxins that affect DNA methylation patterns and identify their potential in the onset and treatment of cancer.

## 1. Introduction

Cancer is a public health concern and a major cause of mortality worldwide. According to the World Health Organization (WHO), the annual global cancer statistics indicated that an estimated 18.1 million new cases and 9.6 million deaths have occurred in the year 2018 [1]. These statistics have increased dramatically over the past few years and are expected to double by the year 2040 [1]. The increasing cancer burden is due to several factors including population growth and aging as well as changes in the prevalence and distribution of cancer risk factors, many of which are associated with social and economic development [1,2]. To date, significant advances have been made in the prevention and treatment of cancer; however, early detection, side effects, drug resistance, treatment costs, and lack of access to health care facilities and palliative care remain major challenges [1].

It is estimated that nearly 50% of all cancers can be avoided by dietary modification [1]. The Western diet coupled with physical inactivity is governed by convenience and comprises of an excessive intake in refined grains, processed meats, sugary desserts, fried foods, and high-fat dairy products that promote an oxidative and inflammatory environment ideal for cancer development [3,4,5,6]. In addition, the consumption of agricultural foods that are contaminated with mycotoxin-producing fungi can alter cellular functions, leading to genetic mutations and the onset of cancer [7,8,9,10,11].

Natural dietary micronutrients and bioactive compounds from fruits, vegetables, and spices have long been investigated, due to their wide availability and fewer side effects, for their potential to prevent and destroy cancer cells [12]. Previously, it was reported that 25–80% of cancer patients used dietary compounds and/or micronutrient supplementation as a therapeutic agent to replenish the body’s nutritional needs after surgery and/or chemotherapy, inhibit tumor growth, and prevent tumor recurrence. Despite this, the exact mechanism by which these compounds prevent and inhibit carcinogenesis remains unclear. Emerging evidence suggests that dietary micronutrients and bioactive molecules exhibit anti-cancer properties by reversing abnormal gene activation and inhibition through epigenetic modifications such as DNA methylation [13,14,15,16,17,18], and these changes in DNA methylation may provide insight into future therapeutic interventions. In this review, we summarize existing literature on natural dietary micronutrients, bioactive compounds, and food-borne mycotoxins that affect DNA methylation patterns and identify its potential in the onset and treatment of cancer.

## 2. Cancer and DNA Methylation

Cancer is defined as the uncontrolled proliferation of abnormal cells that leads to tumor formation with the potential to spread to various organs and tissues within the body. The initiation and progression of cancer is a multi-stage process that occurs from aberrant signaling pathways and is the result of an accumulation in both genetic and epigenetic modifications [19]. Epigenetics refers to inherited modifications that influence gene expression by altering the structure and function of the genome [20]. These reversible modifications occur independently of the DNA sequence and are essential in regulating the cellular phenotype [20]. The most common epigenetic mechanisms include DNA methylation, histone post-translational modifications, RNA methylation, and microRNAs. Among these, DNA methylation is the most dysregulated epigenetic mechanism in cancer and hence, it will be the focus of this review.

DNA methylation is a biochemical process that maintains genomic stability and is often associated with a repressed chromatin structure and gene silencing [21]. DNA methylation plays a major role in the regulation of pluripotency genes, oncogene repression, transposon silencing, genomic imprinting, and X-chromosome inactivation [21]. It controls transcriptional gene silencing during cell proliferation, development, and differentiation, and it is involved in creating distinct cell lineages in adult organisms [20,21].

In eukaryotes, DNA methylation occurs almost exclusively at CpG dinucleotides, where a cytosine nucleotide occurs next to a guanine nucleotide, and it involves the addition of a methyl group from the universal methyl donor, S-adenosylmethionine (SAM), to the carbon-5 position of cytosine residues (Figure 1) [20,21].

This process yields 5-methylcytosine and is catalyzed by DNA methyltransferases (DNMTs) such as DNMT1, DNMT3A, and DNMT3B. DNMT1 is a maintenance methyltransferase that binds specifically to hemi-methylated DNA and is responsible for conserving the DNA methylation pattern from one generation to the next [20]. DNMT3A and DNMT3B are de novo methyltransferases that establish DNA methylation patterns by targeting un-methylated cytosine bases to initiate methylation [20]. In addition to the DNMTs, methylated CpG dinucleotides can also regulate DNA methylation patterns by recruiting methyl-CpG-binding domain (MBD) proteins such as MBD2, which possess demethylase activity [22].

Numerous data indicate that aberrant DNA methylation is a hallmark of cancer [23,24,25]. Global DNA hypomethylation, due to its association with a loss in genome stability and an increase in genetic mutations, has been recognized as a frequent early event in various cancers [26,27,28]. Similarly, the hypermethylation of CpG dinucleotides, mainly those located within the promoter regions of genes, has been observed in tumor tissues compared to surrounding healthy tissues, and it was implicated as a common mechanism by which cancer cells inactivate tumor suppressor genes, as well as escape cell cycle regulatory checkpoints and apoptotic cell death [23,29,30]. Recent evidence also suggests that promoter DNA hypomethylation can lead to cancer by activating genes, including oncogenes, that have been implicated in cellular transformation, invasion, and metastasis [25]. This provides a role for DNA methylation in determining disease severity and metastatic potential.

## 3. The Effect of Natural Dietary Micronutrients and Bioactive Compounds on DNA Methylation in Cancer

Accumulating evidence indicates that the human diet is a source of micronutrients (folate, B vitamins, betaine, choline, and methionine; Figure 2) and bioactive compounds (curcumin, epigallocatechin-3-gallate, genistein, quercetin, resveratrol, and sulforaphane; Figure 2) that act as both chemopreventative and chemotherapeutic agents by modulating the epigenome [13,14,15,16,18].

This can occur through alterations in DNA methylation patterns and is usually the consequence of a direct interaction with the enzymes responsible for establishing DNA methylation marks or by acting as methyl donors and cofactors for DNA methylation reactions [31,32,33,34].

### 3.1. Micronutrients as Methyl Donors

Natural dietary micronutrients are capable of influencing DNA methylation patterns through a biological process known as one-carbon (1C) metabolism [35]. This process comprises of the folate cycle, methionine cycle, and transsulfuration pathway (Figure 3), and it is compartmentalized between the cytoplasm, nucleus, and mitochondria [36].

1C metabolism refers to the transfer of 1C moieties for the synthesis of DNA, amino acids, and phospholipids as well as the methylation of DNA, RNA, and proteins. Micronutrients such as folate, B vitamins, betaine, choline, and methionine serve as methyl donors or cofactors in this multifaceted pathway [37].

Dietary folate, in the form of tetrahydrofolate (THF), is converted to 5,10-methyltetrahydrofolate (5,10-mTHF) by the transfer of a 1C moiety from serine. This process is facilitated by serine hydroxymethyl transferase (SHMT) and its cofactor vitamin B6. 5,10-mTHF is then subsequently reduced to 5-methyl-THF (5-mTHF) by methylene tetrahydrofolate reductase (MTHFR) and its cofactor vitamin B2. 5-mTHF serves as a methyl donor for the re-methylation of homocysteine (Hcy) to methionine; a reaction catalyzed by methionine synthase (MS) and the cofactor vitamin B12 [38].

Similarly, betaine and choline (through its oxidation to betaine) can also provide methyl groups required for the re-methylation of Hcy. This process is facilitated by betaine homocysteine methyl transferase (BHMT) and results in the formation of methionine and dimethylglycine (DMG) [35]. Methionine serves as the precursor to SAM [39]. During methyl transfer, SAM is converted to S-adenosylhomocysteine (SAH), which is subsequently hydrolyzed back to Hcy by adenosylhomocysteinase hydrolase (Ahcy) [40]. If Hcy is not metabolized by either the transsulfuration pathway or re-methylation to methionine, Ahcy will favor SAH biosynthesis rather than hydrolysis [41]. The accumulation of SAH inhibits DNMT activity and hampers DNA methylation processes [40,42].

1C metabolism demonstrates the essential role of micronutrients in cellular pathways, and fluctuations in these nutrients can influence SAM/SAH ratios, thus impacting the epigenome through DNA methylation processes. Several studies have reported the effects of supplementation or deficiency of these nutrients on global or gene-specific alterations in DNA methylation [38]. This can have a long-term effect on gene expression, which could contribute to the pathogenesis or progression of cancer.

### 3.2. Folate

Folate or vitamin B9 is the most well-studied essential micronutrient with regard to DNA methylation [43]. It is obtained through the diet predominantly from green leafy vegetables; however, natural folate lacks stability during storage and processing. For this reason, at least 53 countries have implemented the fortification of dietary staples such as bread and cereals with a synthetic form of folate known as folic acid [44]. Folic acid supplementation is also recommended during pregnancy, as it is vital for fetal growth and development. It is also involved in the synthesis and maintenance of DNA and acts as a methyl donor in 1C reactions for the methylation of DNA, RNA, and proteins [45].

Several in vitro studies utilizing various human cancer cell lines have demonstrated the potential of folic acid to modulate global and gene-specific DNA methylation patterns that can either promote or reduce the risk of cancer development (Table 1).

Interestingly, these studies indicated that the effects of folate deficiency and folic acid supplementation on DNA methylation are cell-, site-, and gene-specific and that the direction of DNA methylation changes may not be the same between global and gene- or site-specific DNA methylation [33,46,47].

Evidence for the role of folic acid supplementation in altering DNA methylation and reducing the risk of carcinogenesis was also demonstrated in in vivo rodent models. In Sprague–Dawley rats, maternal folic acid supplementation (control = 2 mg/kg diet versus supplemented = 5 mg/kg diet) increased global DNA methylation and reduced the risk of colorectal adenocarcinoma in offspring; however, post-weaning folic acid supplementation significantly decreased global DNA methylation in the colon of the offspring at 14 weeks of age and may increase cancer risk [52]. In contrast, maternal and post-weaning folic acid supplementation (control = 2 mg/kg diet versus supplemented = 5 mg/kg diet) increased the risk of mammary tumors in offspring by inducing global DNA hypomethylation and reducing DNMT activity, respectively, in non-neoplastic mammary glands [53]. Folate deficiency (0 mg/kg diet for 4–6 weeks) in weanling Sprague–Dawley rats were shown to selectively induce hepatic *p53* promoter hypomethylation and aberrancies in the *p53* gene that may lead to carcinogenesis in later life [54]. Additionally, in C57BL/6 mice, maternal and post-weaning folate-deficient (0.4 mg/kg diet) diets were shown to modulate colorectal cancer development by inducing *p53* promoter hypomethylation in adults and *adenomatous polyposis coli* (*APC*) promoter hypermethylation in APC^+/min^ offspring, respectively [55].

In humans, controlled feeding trials were undertaken to determine if folate supplementation and depletion affected DNA methylation patterns and cancer outcome. In adults, a mean age of 57.8 years and belonging to the ethnic groups of White, African American, or Hispanic, who had previous cases of colorectal adenomas, the use of 1 mg/day of folic acid increased the methylation of the CpG loci of the proto-oncogenes *estrogen receptor alpha* (*ERα*) and *secreted frizzled-related protein 1* (*SFRP1*) [56]. This may reduce the risk of recurrence of colorectal adenomas; however, Cole et al. showed that 1 mg/day of folic acid for 3 years did not reduce the risk of developing colorectal adenomas in adults aged between 21 and 80 years and belonging to the ethnic groups of White, Black, Hispanic, or Asian [57]. Furthermore, in a long-term follow-up trial, folic acid (400 µg/day) in combination with vitamin B12 (500 µg/day) for 2–3 years increased the risk of developing colorectal carcinomas in Caucasians aged 65 years and over [58]. The daily use of folic acid (800 µg) and vitamin B12 (400 µg) was also shown to increase the risk of developing lung cancer in Norwegians with a mean age of 62.3 years [59]. However, in both studies, the intake of either folate or vitamin B12 was above the recommended dietary intake.

Cravo et al. showed that folate (10 mg/day) supplementation for 6 months was associated with lower DNA methylation levels in the rectal mucosa of patients (aged 49–82 years) with colorectal carcinomas versus those with adenomas [60]. A similar effect was observed in a prospective randomized trial where folate (5 mg/day) supplementation for 6 months increased the extent of DNA methylation in the rectal mucosa of patients (mean age of 62 years) with colonic adenomas [61]. Pufulete et al. showed that folate (400 µg/mL; 10 weeks) supplementation in patients (mean age of 63.9 years) with colorectal adenomas resulted in a 31% and 25% increase in DNA methylation in leukocytes and colonic mucosa, respectively [62].

Folate and DNA methylation was also shown to play a role in breast and lung cancer. Genome-wide methylation analysis of primary breast tumors from 162 women (aged between 30 and 91 years and of Caucasian, African American, Hispanic, or Asian origin) measured 1413 autosomal CpG loci associated with 773 cancer-related genes. This study found that higher folate intake was associated with higher levels of CpG loci methylation, especially in the *interleukin 17 receptor B* (*IL17RB*) CpG locus, which is a gene that is commonly associated with the signaling cascade that promotes cancer cell survival, proliferation, and migration [63,64]. Differential CpG methylation was also found in lung cancer patients. Folate levels were associated with hypermethylation of the tumor suppressors, *Ras-association domain family 1* (*RASSF1A*) and *MTHFR* in lung cancer patients aged 35–70 years [65].

It is unclear as to whether dietary folate or folic acid supplementation results in changes in healthy tissues that can predispose one to cancer. However, it is evident that folate can play a preventative role against cancer. Nonetheless, other factors in combination with folate status such as age, gender, family history, ethnic group, and lifestyle factors (smoking and alcohol consumption) may provoke processes related to cancer risk.

### 3.3. Other B Vitamins

The eight B vitamins are a group of water-soluble heterogeneous substances. Mammals are unable to synthesize B vitamins on their own and hence, they need to be taken up in sufficient quantities from the diet [66]. B vitamins play diverse roles in the human body by acting as cofactors for different enzymatic reactions [66]. As discussed earlier, vitamin B9 (folate) acts as a methyl donor in 1C metabolism, influencing DNA methylation. Vitamins B2, B6, and B12 are essential cofactors in 1C metabolism. Changes in the levels of these vitamins can alter DNA methylation and gene expression and ultimately promote carcinogenesis [37].

Vitamin B2 also known as riboflavin is a cofactor in the folate cycle. Together with MTHFR, it catalyzes the reduction of 5,10-mTHF to 5-mTHF [67]. Changes in vitamin B2 intake can affect the methyl supply and alter DNA methylation patterns [68]. Vitamin B6 or pyridoxine plays a role in both the folate cycle and transsulfuration pathway. In the folate cycle, it acts as a cofactor for SHMT, which is responsible for the formation of 5,10-mTHF from serine and THF; whereas in the transsulfuration pathway, vitamin B6 functions as a cofactor for cystathionine-β-synthase and thus catalyzes the conversion of Hcy to cystathionine [67]. The reduction of vitamin B6 inhibits SHMT, which results in elevated levels of SAH and an inhibition of DNMT activity [69]. Low levels of total vitamin B6 and its circulating form pyridoxal phosphate (PLP) in individuals with colon cancer have been linked to advanced distal colorectal adenomas, and they are weakly associated with early-stage adenomas [70] and esophageal cancer [71].

Vitamin B12, also known as cobalamin, plays an important role in the re-methylation of Hcy to methionine by acting as a cofactor for MS [67]. The relationships among vitamin B12 intake, DNA methylation, and cancer are inconsistent. The methylation score for several tumor suppressor genes in tumors from individuals with head and neck cancer were calculated and compared to dietary intake of vitamin B12. Reduced tumor suppressor gene methylation was observed in individuals with a high intake of the micronutrient compared to individuals with a low vitamin B12 intake [72]. Vitamin B12 reduced promoter methylation of *RASSF1A* in former smokers with lung cancer and enhanced the CpG island methylation in the *MTHFR* promoter of current smokers with lung cancer, suggesting that dietary factors along with lifestyle factors modify methylation patterns [65]. Squamous cell lung cancer [73] and breast cancer [74] tissue showed a localized deficiency of vitamin B12, which may have resulted in the global DNA hypomethylation that was observed. In women positive for human papilloma virus (HPV), vitamin B12 is shown to reduce the risk of cervical intraepithelial neoplasia by maintaining a high degree of methylation on the HPV *E6* gene [75]. Moreover, vitamin B12 is associated with a reduced risk of developing rectal [76] and hepatocellular [77,78,79,80] carcinomas but an increased risk of prostate [81] and esophageal cancer [71]. No association was observed between the intake of vitamin B2, B6, and B12 and the promoter methylation of *E-cadherin*, *p16*^INK4a^, and *retinoic acid receptor beta 2* (*RARβ2*) genes in breast tumor tissues [82].

### 3.4. Betaine and Choline

Betaine and choline are quaternary ammonium compounds with a recommended daily intake of 550 mg and 425 mg for men and women, respectively [83]. Humans can synthesize choline de novo; however, the quantity produced is insufficient to meet the body’s needs [84]. Thus, choline is predominantly obtained from the diet through cruciferous vegetables, eggs, fish, meat, and dairy products [85]. Choline is responsible for maintaining cell membrane integrity, neurotransmission, transmembrane signaling, and lipid synthesis and transport. It also serves as an indirect methyl donor [86]. The oxidation of choline yields betaine, another crucial micronutrient that can also be obtained through the diet mainly from the consumption of sugar beets [87].

Betaine contains three methyl groups and acts as a more efficient methyl donor than choline, which only has one methyl group. The oxidation of choline to betaine occurs via a two-step reaction that is mediated by choline dehydrogenase and betaine aldehyde dehydrogenase. BHMT mediates the transfer of the methyl group from betaine to Hcy to produce methionine, which is in turn converted into SAM. Thus, the supplementation of choline and/or betaine is known to reduce circulating Hcy levels and influence global and gene-specific DNA methylation patterns.

Maternal choline deficiency altered methylation patterns in fetal gene promoter regions. The methylation levels in the promoter region of genes regulating cell cycle (*CDKN3*), calcium binding (*Calb1*), and angiogenesis (*Vegfc* and *Angpt2*) were found to be significantly reduced in fetal hippocampus [88,89,90]; the active transcription of these genes is associated with oncogenesis. Fetal liver tissue collected from Sprague–Dawley rats fed either choline-deficient or supplemented diets also revealed surprising results. The promoter methylation of *insulin growth factor 2* (*IGF2*), a gene often overexpressed in various cancers, was significantly higher in fetal liver tissue from dams fed choline-deficient diets compared to control tissue, and no significant changes were found in choline-supplemented dams. The hypermethylation of *IGF2* promoter regions was attributed to the concomitant hypomethylation of the *DNMT1* promoter and its increased expression [91].

In contrast, rats fed choline-deficient diets displayed increased levels of SAH, reduced hepatic DNMT activity, and DNA methylation, as well as an increased incidence of developing hepatocellular carcinoma [92,93,94]. Additionally, low choline diets affect the promoter methylation of oncogenes and tumor suppressor genes. Hypomethylation in the promoter region of the oncogene *c-myc* was found in hepatocellular carcinoma tissue of choline-deficient rats, whereas the promoters of some tumor suppressor genes (*p53*, *p16^INK4a^*, *PtprO*, *Cdh1*, and *Cx26*) were hypermethylated [94,95,96]. Furthermore, deletion of the BHMT gene resulted in the spontaneous development of pre-neoplastic foci in the liver, supporting the protective role of choline and betaine against cancer [96].

### 3.5. Methionine

Methionine is one of the eight essential amino acids that can be obtained from protein-rich sources such as meat, fish, eggs, soy, and dairy [97]. Unlike other essential amino acids that must be consumed, methionine can be recycled via the re-methylation of Hcy [98]. It plays a key role in the synthesis of other amino acids and proteins [97,99]. Being a precursor to SAM, methionine is a major driver of DNA methylation, and fluctuations in methionine can significantly alter the methylation status of DNA [41].

Although methionine is essential in DNA methylation, there is a lack of studies that have investigated the effect of methionine on DNA methylation and its relationship with cancer. In a prospective study by Vineis et al., the effect of methionine on the methylation patterns in candidate genes was investigated in lung cancer patients and a set of controls. Methionine was associated exclusively with decreasing methylation levels of *p16^INK4a^, RASSF1A*, and *MTHFR* in former smokers and *RASSF1A* and *GSTP1* in current smokers [65]. *RASSF1A* and *p16^INK4a^* are well-known tumor suppressors that are frequently hypermethylated in lung cancer; thus, methionine supplementation may have a protective role in lung cancer patients [100,101]. In contrast, Bassett et al. and Fanidi et al. found no association between methionine intake and lung cancer risk [102,103]. The methylation status of tumor-suppresser genes—namely, *E-cadherin, p16^INK4a^*, and *RARβ2*—were investigated in breast cancer patients. Methionine did not affect the promoter methylation of these genes, and no association between methionine and breast cancer was found [82,104]. However, a dose–response meta-analysis found that the risk of breast cancer was reduced by 4% for every 1 g/day increment in dietary methionine intake [105]. Although no direct link has been established between DNA methylation and methionine, methionine supplementation also seems to play a protective role in colorectal cancer [106,107,108,109]. Methionine deficiency is associated with increased risk [108], whereas high intake reduces the risk of colorectal cancer [109] and colorectal adenomas greater than 1 cm [110]. The lack of correlation between methionine intake and DNA methylation may be due to the cyclic nature of the re-methylation cycle; high levels of methionine prevent the re-methylation of Hcy, causing SAH to accumulate and inhibit DNMT activity, thus reducing DNA methylation [41].

Interestingly, methionine restriction is being investigated as an anti-cancer strategy (thoroughly reviewed by Chaturvedi et al.) [111]. Due to the methionine cycle, normal cells are relatively resistant to exogenous methionine restriction; however, several human cancer cell lines and primary tumors are unable to grow when methionine is replaced with its precursor Hcy, making cancer cells methionine-dependent [112,113]. Preclinical studies in rodent models have revealed that methionine restriction is effective against carcinomas [114] and inhibits the development of colonic tumors [115] and prostatic intraepithelial neoplasia [116]. Clinical studies using methionine restriction are limited and more complicated to control than rodent studies; however, dietary methionine restriction together with the use of chloroethyl nitrosourea chemotherapy in patients with recurrent glioma or metastatic melanoma reduced O-6-methylguanine-DNA methyltransferase (MGMT) activity, which is a main mechanism of nitrosourea resistance [117].

### 3.6. Curcumin

Curcumin, scientifically known as diferuloylmethane, is a yellow polyphenolic compound found in the perennial herb, *Curcuma longa* [118]. It contains 80% curcuminoid complex, 17% dimethoxycurcumin, and 3% bisdemethoxy-curcumin [119]. This widely used spice and food-coloring agent exhibits potent anti-oxidant [120], anti-inflammatory [121], and anti-cancer [14,118] properties both in human cancer cell lines and animal models. The anti-cancer effect of curcumin is mainly attributed to its ability to induce apoptosis in cancer cells without cytotoxic effects on healthy cells. Thus far, several human in vitro studies have investigated the effects of curcumin on DNA methylation [14,31,118]. Curcumin was shown to induce DNA hypomethylation in several different human tumor cell lines by inhibiting the activities of DNMT1 and DNMT3B [31,118,122,123,124]. Molecular docking studies indicated that curcumin covalently blocks DNMT1 catalytic activity, thus providing mechanistic insight into curcumin-induced DNA hypomethylation [31]. Furthermore, curcumin-induced promoter DNA hypomethylation was associated with the reactivation of tumor suppressor genes such as *p15^INK4B^* in acute myeloid leukemia [122,125] and *RARβ* in lung cancer and cervical cancer cell lines [123,124]. In breast cancer (MCF7 and MDA-MB-231) cells, curcumin (10–20 µM; 72 h) was shown to decrease cell proliferation by inducing a promoter DNA hypomethylation-mediated reactivation of the *breast cancer 1* (*BRCA1*) tumor suppressor and DNA repair gene and the *RASSF1A* tumor suppressor gene, whilst inducing promoter DNA hypermethylation and suppressing the expression of the proto-oncogene *gamma-synuclein* (*SNCG*) [14,126]. Curcumin (10 µM, 72 h) treatment in MCF7 cells was shown to reverse promoter hypermethylation and reactivate the expression of *glutathione S-transferase pi 1* (*GSTP1*) protein [127]. Similarly, curcumin induced promoter DNA hypomethylation of *NRF2* [128] and *Neurog1* (a cancer methylation marker) [129] in mice and human prostate cancer cell lines, respectively. Another study revealed that through global DNA hypermethylation, curcumin (0.01–100 µM; 24 h) decreased the expression of the *argyrophilic nucleolar protein* (*AgNOR*), which is a protein that is highly expressed in malignant cells compared to normal cells and reflects the rapidity of cancer cell proliferation [130]. Curcumin (20 µM; 48 h) also significantly induced apoptosis and decreased HSC-T6 and primary human stellate cell viability via promoter hypomethylation and upregulation of the *phosphatase and tensin homologue* (*PTEN*) [131]. The promoter hypermethylation and downregulation of the NFκB regulated gene *cyclooxygenase 2* (*COX2*) by curcumin prevents inflammatory-driven colon cancer [132]. In contrast, some studies have observed no change in global and gene-specific DNA methylation by curcumin in both leukemia and colorectal cancer cell lines [118,133,134].

### 3.7. Quercetin

Quercetin, a flavonoid and polyphenolic compound found in various fruits and vegetables, displays anti-oxidant and anti-inflammatory effects by scavenging free radicals and chelating transition metal ions [135]. As a result, quercetin may aid in preventing and treating certain diseases such as cancer. Studies on the anti-cancer effects of quercetin indicate that quercetin induced apoptosis and cell growth inhibition in various human cancer cell lines and animal models [136,137,138]. It has been shown that quercetin treatment caused cell cycle arrest at the G2/M phase in human breast and laryngeal cancer cells [139,140], and at the G1 phase in colon cancer cells and leukemic cells [141,142]. In human bladder cancer cell lines, quercetin (0–100 µM; 24–48 h) was shown to decrease the DNA methylation levels of *p16^INK4a^* and *RASSF1A*, leading to cell cycle arrest and apoptosis [143]. Quercetin (25 and 50 µM; 24–48 h) interacts with the catalytic components of DNMTs and inhibits DNMT activity, leading to global DNA hypomethylation in human cervical cancer (HELA) cells [32]. This was shown to occur in a dose- and time-dependent manner. Quercetin also dose-dependently decreased the promoter methylation of several tumor suppressor genes with restoration of their expression [32]. The expression of DNMT1 and DNMT3A was also significantly reduced by quercetin (50 and 75 µmol/L; 48 and 72 h) in the HL60 and U937 leukemia cell lines, with a corresponding decrease in the promoter methylation of the apoptosis-related genes *BCL2L11* and *DAPK1* [144]. In addition, hypermethylation of the *p16^INK4a^* gene was reversed by quercetin (5–50 µM; 120 h) followed by a concentration-dependent restoration in its expression [145].

### 3.8. Resveratrol

Resveratrol (3,5,4′-trihydroxy-*trans*-stilbene) is a stilbenoid compound derived from peanuts, grapes, and berries [146]. It is a phytoalexin that was proved to have both anti-oxidant and pro-oxidant properties, leading to its recognition as a chemopreventative agent [147]. Resveratrol also exhibits effective anti-cancer activity through epigenetic mechanisms. Genome-wide methylation analysis showed that resveratrol (100 µM) treatment for 24 and 48 h decreased gene promoter hypermethylation and increased DNA hypomethylation in breast cancer (MDA-MB-231) cells [148]. The inhibitory effect of resveratrol against DNMTs in cancer cells is well documented [149,150,151,152]. Resveratrol (15 µM) was shown to decrease the enzymatic activity and mRNA expression levels of DNMT1, DNMT3A, and DNMT3B in MDA-MB-157 and HCC1806 breast cancer cell lines, which led to a significant decrease in 5-methylcytosine levels and an overall reduction in global DNA methylation patterns [151,153]. This decrease in DNA methylation led to the reactivation and increase in the expression of *ERα* and provided insight into hormone-directed anti-cancer therapies [153]. Similarly, resveratrol was shown to inhibit DNMT1 activity and expression at the *BRCA1* promoter, leading to promoter hypomethylation and the reactivation of *BRCA1* in breast cancer cells [152,154]. A decrease in DNMT1 expression by resveratrol (14 µM; 72 h) was also accompanied with a decreased *PTEN* promoter methylation and increased *PTEN* expression levels in the MCF7 breast cancer cell line [155]. The anti-cancer effects of resveratrol were also observed in non-small cell lung cancer (A549) cells, where through suppressed DNMT1 expression, resveratrol (20–100 µM; 72 h) reduced promoter methylation and upregulated the expression of the *zinc finger protein 36* (*ZFP36*), leading to an inhibition in cell migration and proliferation [16]. In vivo studies indicate that resveratrol (25 mg/kg/day in a rodent model of estrogen-dependent mammary carcinoma) decreased DNMT3B expression in tumor tissues versus normal mammary tissue; however, no effect was observed on DNMT1 expression [150]. In contrast, resveratrol (15 µM) treatment for 9 days increased DNMT3B activity that led to the hypermethylation and silencing of the *MAML2* (*mastermind-like protein 2*; co-activator of the oncogenic NOTCH signaling pathway) gene in MCF10CA1h and MCF10CA1a human breast cancer cell lines [156]. Resveratrol also modulates DNA methylation patterns by altering the levels of SAM and SAH [146]. The expression of the methyl-DNA-binding proteins, MeCp2 and MBD2, were also significantly reduced by resveratrol in breast cancer cells [157]. In glioma cells, resveratrol increased sensitivity to temozolomide-induced apoptosis by downregulating the activity and expression of the DNA repair protein, MGMT. This was shown to occur via inactivation of the Wnt and NF-κB signaling pathways, which is indicative of yet another DNA methylation inhibiting activity of resveratrol [158,159].

### 3.9. Sulforaphane

Sulforaphane (SFN), an isothiocyanate derived from cruciferous vegetables, has been demonstrated to prevent and reduce tumor incidence and progression in various models of breast, cervical, colon, and lung cancers [17,160,161,162]. Studies on the mechanisms underlying the anti-cancer activities of SFN indicate that its regulatory effects on the tumor cell cycle, apoptosis, and angiogenesis are mediated by an epigenetic modulation of essential cell signaling pathways and genes. Previous studies show that SFN decreased DNMT1, DNMT3A, and DNMT3B expression, leading to global DNA hypomethylation in human breast, prostate, and cervical cancer cell lines [13,163,164,165,166]. Furthermore, the inhibitory effects of SFN on DNMTs were shown to reactivate the expression of silenced genes in cancer cells via promoter demethylation. An experiment with LnCap prostate cancer cells showed that SFN (30 µM; 24 h) induced promoter demethylation and expression of the cell cycle regulatory gene *cyclin D2*, which correlated with an increase in cell cycle arrest and cell death [164]. SFN (1.0 and 2.5 µM) treatment of prostate cancer cells from TRAMP mice (Transgenic Adenocarcinoma of the Mouse Prostrate) and TRAMP C1 cells indicated that SFN plays a role in detoxification by decreasing *NRF2* promoter methylation. This *NRF2* promoter hypomethylation occurred as a result of reduced DNMT expression and was associated with an increase in *NRF2* expression as well as an increase in the mRNA and protein expression of its downstream target NAD(P)H quinone dehydrogenase 1 (NQO1) [167]. In human cervical cancer (HELA) cell line, SFN (2.5 µM) caused cell cycle arrest and apoptosis by decreasing the expression and enzymatic activity of DNMT3B, which led to promoter hypomethylation and the upregulation of tumor suppressor genes (*RARβ*, *CDH1*, *DAPK1*, and *GSTP1*) and pro-apoptotic proteins (Bax) [13]. SFN (25 µmol/L) also induced *NRF2* promoter hypomethylation and expression in human colon cancer (Caco-2) cells; however, the decrease in *NRF2* promoter methylation was attributed to a decrease in the protein expression and activity of DNMT1 [168]. A similar effect was observed in MCF7 and MDA-MB-231 breast cancer cell lines where exposure to 10 µM SFN reduced DNMT1 expression and elevated the expression of *PTEN* and *RARβ2* via promoter demethylation, thus enhancing cell growth arrest and apoptosis [169]. SFN (10 µM; 48 h) was also found to modulate estrogen-induced DNA methylation of the *catechol-O-methyltransferase* (*COMT*) gene, which functions as a gatekeeper to prevent DNA damage and tumorigenesis during estrogen metabolism [170]. Another study indicated that SFN (10 µM) suppressed DNMT1 and DNMT3A expression, which in turn led to site-specific CpG demethylation and downregulation of the *human telomerase reverse transcriptase* (*hTERT*) gene, which is a catalytic subunit of telomerase. This decrease in *hTERT* expression facilitated the induction of apoptosis in human MCF7 and MDA-MB-231 breast cancer cell lines [163]. A similar result was observed in LnCap and DU145 prostate cancer cell lines, where SFN (15 µM) modulated telomerase activity by decreasing the promoter methylation and expression of *hTERT* [171]. In nasopharyngeal primary carcinoma cells, SFN inhibited tumor growth via the downregulation of DNMT1 and re-expression of the Wnt inhibitory factor 1 (WIF1) [172]. In contrast, SFN (2.0–32.0 µM) treatment in human hepatocellular carcinoma (HepG2) cells induced cell cycle arrest and apoptosis through the hypermethylation of genes such as *E2F3*, *THAP1*, and *ANKHD1*, which are involved in cell cycle progression and cell proliferation [173]. Interestingly, through alterations in promoter DNA methylation, the consumption of SFN in combination with anti-cancer drugs such as Withaferin A and Clofarabine exhibited a more potent and favorable effect on cell growth arrest and apoptosis in MCF7 and MDA-MB-231 breast cancer cell lines [169,174]. SFN in combination with the DNA methyltransferase inhibitor 5-aza-2-deoxycytidine also showed effective inhibition against mouse melanoma B16F10 cell growth [175]. SFN (20 µM) was also shown to enhance the anti-tumor effects of cisplatin in nasopharyngeal primary carcinoma cells [172].

### 3.10. Genistein

Genistein (5,7-dihydroxy-3-(4-hydroxyphenyl)chromen-4-one), an isoflavonoid and phytoestrogen commonly found in soybeans and soy-based products, is well-known for its anti-angiogenic and anti-cancer properties [176,177,178]. The DNA methylating capabilities of genistein in cancer cells have been documented [15,179,180]. In breast cancer (MCF-7 and MDA-MB-231) cell lines, genistein (60 and 100 µM) was shown to decrease global DNA methylation by inhibiting DNMT activity and DNMT1 expression. The decrease in DNMT activity further decreased the promoter methylation of multiple tumor suppressor genes such as *ataxia telangiectasia mutated* (*ATM*), *APC*, *PTEN*, and *mammary serpin peptidase inhibitor* (*SERPINB5*), leading to an upregulation in their expressions and a decrease in cell viability [179]. The decrease in *BRCA1* gene promoter methylation and expression by genistein in breast cancer cells was also attributed to a decrease in *DNMT1* expression both in vitro (MCF7, UACC3199, and HCC38 cell lines) and in vivo (BRCA1 F22/24 mice) [181,182]. Genistein also induced promoter hypomethylation and reactivated the expression of tumor suppressor genes *GSTP1*, *EPHB2*, *p21*, *RARβ*, *p16^INK4a^*, and *MGMT* in breast cancer, kidney cancer, esophageal squamous cell carcinoma, and prostate cancer cell lines, leading to cell cycle arrest and cell death [15,183,184,185]. The expression of *E-cadherin*, *DAPK1*, *RARβ*, *p16^INK4a^*, *MGMT*, *FHIT*, *RUNX3*, *CDH1*, *PTEN*, and *SOC51* was also reactivated through genistein-induced promoter hypomethylation in cervical cancer cells [186,187]. Genistein induced promoter hypomethylation and increased the expression of the tumor suppressor gene *BTG3* in human prostate (LnCap and PC3) cancer [188], renal (A498 and ACHN) carcinoma [189], and leukemia (MOLT17, MOLT4, and Jurkat) [190] cell lines, leading to cell cycle arrest and a decrease in cell proliferation. A decrease in cell proliferation and an increase in apoptosis through promoter demethylation and the upregulation of *CDKN2a* was also observed by genistein (25–100 µM) in kidney cancer cell lines [18]. In the HT29 colon cancer cell line, genistein (10, 20, and 60 µmol/L; 72 h) inhibited cell invasion and migration by inducing demethylation and recovering the activity of WIF1 [191]. Contrastingly, in Sprague–Dawley rats, genistein (140 mg/kg) was found to decrease cell proliferation and induce the apoptosis of colon cancer cells through promoter hypermethylation and inactivation of the Wnt signaling activators, *Sfrp2*, *Sfrp5*, and *Wnt5a* [180]. However, another study using SW1116 colon cancer cells showed that genistein (75 µmol/L) treatment for 4 days inhibited cell proliferation by decreasing *Wnt5a* CpG island methylation and upregulating its expression [192]. Genistein (2–20 µmol/L) inhibited the proliferation of human esophageal squamous cell carcinoma (KYSE 510) cell line by reversing DNA hypermethylation and reactivating the expression of *RARβ*, *p16^INK4a^*, and *MGMT*. This reversal of *RARβ* promoter hypermethylation and increase in its expression was also observed by genistein in KYSE 150 cell line and prostate cancer (LNCaP and PC3) cell lines [15]. Genistein in combination with 5-aza-2-deoxycytidine enhanced the expression of *RARβ* and *p16^INK4a^* and inhibited the growth of the KYSE 510 cell line [15].

### 3.11. Epigallocatechin-3-Gallate

Green tea derived from the tea plant, *Camellia sinensis*, possesses significant health benefits due to an abundance of monomeric catechins such as epicatechin (EC), epicatechin-3-gallate (ECG), epigallocatechin (EGC), and epigallocatechin-3-gallate (EGCG). Of these, EGCG is the most active ingredient of green tea polyphenols and is shown to have anti-cancer activities through alterations in epigenetic modifications [193,194,195,196]. EGCG (5–50 µM) was found to induce global DNA hypomethylation by inhibiting DNMT activity [194]; and the inhibition in DNMT activity is a result of direct blockage of the active site of DNMTs as well as a reduction in the levels of SAM and SAH [194,197]. EGCG (50 µM) also disrupts global DNA methylation by altering folic acid metabolism [198]. EGCG-mediated inhibition of DNMTs has been linked with promoter demethylation and the re-expression of several tumor suppressor genes such as *p21*, *p16^INK4a^*, *RARβ*, *MGMT*, and *human mutL homologue 1* (*hMLH1*) in various cancer cell lines [193,194,198,199]. Similarly, the topical application of EGCG (1 mg/cm^2^ of skin area for 30 weeks) restored global DNA hypomethylation patterns and provided protection against ultraviolet-B-induced carcinogenesis in the SKH-1 hairless mouse model [200]. Another study indicated that treatment of the A431 skin cancer cell line with EGCG (5–20 µg/mL; 3 and 6 days) inhibited cell proliferation and induced apoptosis via the inhibition of DNMT expression and activity, global DNA hypomethylation, and the re-expression of the key tumor suppressor genes *p21* and *p16^INK4a^* [193]. In the cervical cancer (HELA) cell line, EGCG (25 µM) suppressed DNMT3B activity and expression, leading to promoter hypomethylation and the reactivation of *RARβ*, *CDH1*, and *DAPK1* [195]. EGCG inhibited DNA methylation as well as the catalytic activities of the cytochrome P450 enzymes, CYP1A, CYP2B1, and CYP2E1 preventing lung tumorigenesis in A/J mice (an inbred albino strain of mice commonly used in cancer studies) [201]. In human lung cancer (A549 and H460) cell lines, EGCG (0–50 μM; 72 h)-induced promoter demethylation and the restoration of *WIF1* expression downregulated the Wnt signaling pathway but did not inhibit cell proliferation [196]. A decrease in cell proliferation was only observed for EGCG concentrations ranging from 50 to 200 µM. EGCG (50–350 µM; 72 h) caused cell cycle arrest and decreased cell viability by inhibiting DNMT activity and inducing *RXRα* promoter hypomethylation and expression in human colon cancer cells [202]. EGCG (25–200 mg/L; 96 h) was also found to inhibit cell growth and induce apoptosis in esophageal cancer (ECa109) cell line through demethylation and reactivation of the *p16^INK4a^* gene [203]. In breast cancer (MCF7 and MDA-MB-231) cell lines, EGCG (20 μM; 48 h) reduced DNMT expression and activity, which in turn decreased the methylation status of *SCUBE2* (*signal peptide-CUB (complement protein C1r/C1s, Uegf, and Bmp1)-EGF (epidermal growth factor) domain-containing protein 2*). This decrease in *SCUBE2* methylation enhanced *SCUBE2* expression and suppressed cell migration and invasion [204]. Similarly, in HSC3 and SCC9 oral carcinoma cell lines, EGCG (5–50 µM; 6 days)-induced inhibition of tumor invasion, angiogenesis, and metastasis was correlated with partial demethylation and upregulation of the tumor suppressor, *RECK* [205]. EGCG (40 µM) treatment of MCF7 and MDA-MB-231 breast cancer cell lines for 3, 6, 9, and 12 days displayed a decrease in cell proliferation and increase in apoptosis; this was thought to occur via *hTERT* promoter hypomethylation and inhibition and was the consequence of a decrease in DNMT activity [206].

### 3.12. Combinational Effects of Dietary Compounds on DNA Methylation in Cancer

Although the use of a single bioactive dietary compound has shown promise against the growth and risk of cancer as discussed above, evidence suggests that the combination of two or more bioactive dietary compounds can target multiple pathways to induce a more potent effect on DNA methylation and cancer. In one study, the treatment of prostate cancer (PC3 and DU145) cell lines with curcumin (5.0 µM) and quercetin (5.0 µM) for 48 h enhanced the decrease in cell proliferation and apoptosis by inhibiting DNMT activity and inducing global DNA hypomethylation as well as *androgen receptor* (*AR*) promoter hypomethylation and reactivation [207]. Exposure of the MDA-MB-231 cell line to a mixture of EGCG (20.0 µg/mL) and SFN (5.0 µM) for 72 h also enhanced cell growth inhibition and apoptosis by inducing global DNA hypomethylation as well as *ERα* promoter hypomethylation and expression [208]. Similarly, a 48 h treatment of the MCF7 cell line with a mixture of genistein (1 µM) and SFN (10 µM) synergistically increased CpG-site specific DNA methylation at the *MB-COMT* distal promoter and expression, leading to a decrease in cell proliferation, as measured by reduced BrdU (bromodeoxyuridine) incorporation [170].

### 3.13. Clinical Trials with Bioactive Dietary Compounds and DNA Methylation in Cancer

Due to its promising role as both chemopreventative and chemotherapeutic agents, several bioactive dietary compounds have entered clinical trials. In a randomized clinical trial on 10 male Norwegian patients aged between 55 and 69 years who received either 30 mg genistein or placebo capsules daily for 3–6 weeks before prostatectomy, whole genome methylation and expression profiling identified several differentially methylated sites and expressed genes between placebo and genistein groups. The majority of these genes were found to be involved in cell proliferation, thus highlighting the effects of genistein on global changes in gene expression in prostate cancer [209]. In another randomized clinical trial, 34 healthy premenopausal women (aged between 19 and 54 years) fed isoflavones including genistein at 40 or 140 mg daily throughout one menstrual cycle showed hypermethylation of 5 key cancer-related genes (*p16^INK4a^*, *RASSF1A*, *RARβ2*, *ER*, and *CCND2*) [210]. In contrast, resveratrol (5 or 50 mg) administered twice daily for 12 weeks to 39 women (median age of 59.5 years in the 5 mg group versus a median age of 54 years in the 50 mg group) with increased breast cancer risk displayed chemopreventative properties by inducing *RASSF-1α* demethylation and re-expression [211].

## 4. Mycotoxins

The consumption of foods contaminated by mycotoxins is a major health hazard that can lead to the onset and progression of cancer. Each year, approximately 25% of the global food and feed output is contaminated by mycotoxins, threatening food security [212]. Mycotoxins are toxic secondary metabolites produced by fungi that parasitize agricultural crops. Mycotoxins are produced in response to several environmental factors such as warm humid conditions, and constant exposure to high levels of mycotoxins is common in areas where there are no regulations that protect the food intake of the populace [212]. Mycotoxin exposure is also common in areas with poor food-handling and storage methods [212].

Some of the main food-borne mycotoxins—fusaric acid, fumonisin B_1_, deoxynivalenol, T-2 toxin, zearalenone, ochratoxin A, and aflatoxin B_1_ (Figure 4)—have been shown to induce a change in the DNA methylation profile in human cell lines and animal models [213,214,215,216,217,218,219].

The effects of these mycotoxins on DNA methylation are presented in detail hereafter.

### 4.1. Fusaric Acid

The metal chelating agent, fusaric acid (FA), is a secondary metabolite produced by the *Fusarium* species. FA contaminates maize and cereal grains, inducing low to moderate toxicity in animals [220]. This chelating agent binds to divalent cations, obstructing their activity in biological processes, including blood coagulation [221], bone ossification [222], hypotension [223,224], and notochord formation [225]. FA has been implicated in oxidative stress [226], mitochondrial dysfunction [226], altered membrane permeability [227], DNA damage [228], cell death [226,229], and immunotoxicity [229]. Additionally, FA toxicity may be attributed to synergistic interactions with other co-occurring mycotoxins [220].

To further understand the effects of FA and identify an alternative mechanism of FA-induced toxicity, Ghazi et al. investigated the epigenetic effects of FA (25, 50, 104, and 150 µg/mL) in the HepG2 cell line. This study indicated that FA decreased 5-methylcytosine DNA content, which is indicative of global DNA hypomethylation in the HepG2 cell line. The decrease in global DNA methylation occurred as a result of the decrease in DNMT1, DNMT3A, and DNMT3B expression as well as an increase in MBD2 expression [213]. Furthermore, FA was shown to decrease DNMT1, DNMT3A, and DNMT3B expressions by inducing promoter hypermethylation of the *DNMTs*. Similarly, the FA-induced increase in MBD2 expression was shown to occur due to *MBD2* promoter hypomethylation [213]. This study indicated that FA may induce genotoxicity and cytotoxicity in the HepG2 cell line by altering their DNA methylation profile.

### 4.2. Fumonisin B_1_

Fumonisin B_1_ (FB_1_) is produced by several *Fusarium* species and is the most common fungal contaminant of maize and other agricultural crops [230,231,232]. It has been classified as a type 2B carcinogen by the International Agency for Research on Cancer (IARC) [233]. Due to its structural similarity with sphingoid bases, FB_1_ exerts its toxic effects by inhibiting ceramide synthase (*N*-acetyltransferase), which disrupts sphingolipid biosynthesis. This results in the accumulation of sphingoid bases, affecting cellular processes such as cell differentiation, cell proliferation, cell death, and oxidative stress [234]. Due to this disruption of sphingolipid biosynthesis, FB_1_ has been proved to induce tumor proliferation in the liver and kidney [235]. FB_1_ contamination in corn (223.6 μg/g) and rice (21.6 μg/g) was associated with an increased risk of esophageal cancer in the Golestan Province of Northeastern Iran [236]. The consumption of corn contaminated by FB_1_ at concentrations of 8.67 µg/kg body weight per day in the Transkei region of South Africa and 18–155 ppm in the Cixian and Linxian counties of the People’s Republic of China has also been correlated with a high incidence of esophageal cancer [237,238]. The approximate daily intake of FB_1_ may range from 12 to 140 µg; however, in South Africa, where maize is a staple food, the average daily intake can include up to 2500 µg [239]. FB_1_ may be considered an epigenetic carcinogen in risk assessment processes [240,241,242], and various studies have revealed that folate deficiency due to FB_1_ leads to the disruption of DNA methylation [243,244,245].

A study by Mobio et al. indicated that a concentration of 9–18 µM of FB_1_ for 24 h induced cell cycle arrest by causing DNA hypermethylation in the rat C6 glioma cell line; however, concentrations exceeding 18 µM failed to induce DNA hypermethylation [246]. More recently, Chuturgoon et al. investigated the effect of FB_1_ on DNA methylation in the HepG2 cell line. They documented that FB_1_ (200 µM) induces global DNA hypomethylation in the HepG2 cell line via a mechanism that alters the DNA methyltransferase/demethylase balance, thereby disrupting the structural integrity of DNA [214]. FB_1_ induced hypomethylation through a significant decrease in *DNMT1*, *DNMT3A*, and *DNMT3B* expression, with a significant increase in *MBD2* expression [214]. The data presented by Chuturgoon et al. implicate FB_1_ as a hepatocarcinogen and propose that FB_1_ plays a role in cellular transformation and invasion in HepG2 cells.

Demirel et al. determined the effect of increasing concentrations of FB_1_ from 1 to 50 µM for 24 h on cell viability and on the CpG methylation levels of tumor suppressor genes and oncogenes in liver and kidney (Clone 9 and NRK-52E) cell lines. In this study, aberrant CpG promoter methylation was not detected in the cell adhesion molecule *E-cadherin* or the cyclin-dependent kinase inhibitor *p15* in both Clone 9 and NRK-52E cell lines. The cyclin D-dependent kinase inhibitor *p16* was found to be methylated in NRK-52E cells but un-methylated in Clone 9 cells. In Clone 9 cells, the level of CpG promoter methylation was observed in the oncogene *c-myc* albeit only at FB_1_ concentrations of 10 and 50 µM. The methylation of the *c-myc* promoter was decreased for all FB_1_ concentrations in NRK-52E cells. CpG promoter methylation of the *Von Hippel-Lindau* (*VHL*) gene was detected in Clone 9 cells at FB_1_ concentrations of 10, 25, and 50 µM and in NRK-52E cells at concentrations of 10 and 50 µM. Despite the changes in the DNA methylation profile, FB_1_ had no obvious decrease in cell viability [247].

### 4.3. Ochratoxin A

Ochratoxin A (OTA) is produced by various *Aspergillus* and *Penicillium* species [248]. It contaminates a variety of food commodities, including dried fruit, green coffee, cocoa, cereals, and meat products. OTA is distributed worldwide and has a wide range of toxicological effects, including hepatotoxicity, nephrotoxicity, immunotoxicity, neurotoxicity, teratogenicity, and genotoxicity [249]. After absorption in the small intestine, OTA enters the bloodstream, where it has the longest half-life in humans [250] and is distributed to different tissues, with its primary target being the kidney [249,251,252]. OTA has been shown to induce kidney damage and renal carcinoma in mice and rats [253,254], and it has been classified as a group 2B carcinogen by the IARC [255]. Several OTA mechanisms of action have been proposed, as the mechanism of OTA cytotoxicity differs from one tissue to another.

The effect of OTA on DNA methylation levels in human cells has become a topic of interest. Consequently, the level of DNA methylation has become a marker of many diseases [256]. To improve current knowledge on OTA and DNA methylation, Giromini et al. estimated the modifications in the DNA methylation level of BME-UV1 and MDCK cell lines by an OTA concentration of 1.25 µg/mL for 24 h; both BME-UV1 and MDCK cell lines showed no change in global DNA methylation patterns [257]. Similarly, Ozden et al. showed no change in global DNA methylation in the kidneys of male rats following 14, 28, and 90 days of an OTA intake of 21, 70, or 210 μg/kg. However, microarray analysis of DNA methylation at 90 days showed that OTA induced differential DNA methylation levels in the promoters of genes involved in protein kinase activity, phosphate metabolism, and mTOR (mammalian target of rapamycin) signaling, suggesting a role of OTA in carcinogenicity [258]. These methylation changes included the hypermethylation of *Tbc1d5*, *Arap2*, *Ano6*, *Cul2*, and *Dlg2* gene promoters and the hypomethylation of *Cpne4*, *Pdpk1*, *Spop*, *Ogdh*, *Dock3*, and *Rptor* gene promoters [258].

Another study by Li et al. investigated the effects of low (70 µg/kg) and high (210 µg/kg diet) OTA intake on global DNA methylation in rat kidneys after 4, 13, and 26 weeks of consumption. After 4 weeks, global DNA methylation was reduced in rat kidneys exposed to a high dose of OTA, while low-dose OTA treatment showed no change in methylation [215]. Remarkably, after 13 weeks, the kidneys of both low and high-dose OTA-treated rats showed a significant increase in global DNA methylation [215]. However, after 26 weeks, the OTA-exposed kidney showed no change in global DNA methylation [215]. Upon examination of DNMT expressions in OTA-exposed kidneys after 13 weeks, it was found that *DNMT1* and *DNMT3B* transcripts were increased by OTA, while *DNMT3A* was significantly decreased. Li et al. also determined the effects of OTA on DNA methylation in the promoter regions of *E-cadherin* and *N-cadherin*; the methylation status of *E-cadherin* and *N-cadherin* promoters after 4 weeks were unaffected but showed hypermethylation after 13 weeks [215]. The promoter hypermethylation of *E-cadherin* and *N-cadherin* is often associated with a loss in its expression and has been previously observed in human tumors. Furthermore, Zheng et al. demonstrated that the toxic effect of OTA (25 µM) in the HepG2 cell line was a result of global DNA hypomethylation, as indicated by the significant decrease in the DNA 5-methylcytosine content [259]. Since global DNA hypomethylation and promoter hypermethylation are hallmarks of cancer, OTA-induced changes in DNA methylation may provide a mechanism for controlling the expression of genes involved in carcinogenesis.

### 4.4. Aflatoxin B_1_

Aflatoxin B_1_ (AFB_1_) is a group 1 carcinogen produced by *Aspergillus flavus* and *Aspergillus parasiticus* [260]. AFB_1_ is widely distributed in foods such as groundnuts, milk, rice, sorghum, maize, and oils [261,262], which form part of the staple diet. Once ingested, AFB_1_ is metabolized in the liver by cytochrome P450 enzymes to produce aflatoxin-8,9-exo-epoxide, which is an unstable reactive intermediate that is capable of binding to DNA, RNA, and proteins to inhibit their functions [263].

Hepatocellular carcinoma is the most common cancer worldwide. Since the transversion mutation (G→T) on codon 249 of the *p53* tumor suppressor gene has been associated with AFB_1_ intake [264], extensive research has been conducted on the relationship between AFB_1_ concentration and the incidence of hepatocellular carcinoma. Zhang et al. investigated the DNA methylation of *RASSF1A* and *p16^INK4a^* promoters in 83 hepatocellular carcinoma tissue samples from Taiwan. They found a correlation between the *RASSF1A* and *p16^INK4a^* methylation statuses and the AFB_1_-DNA adduct levels [265]. This study concluded that exposure to AFB_1_ may induce carcinogenesis by altering the DNA methylation status and the expression of genes associated with cancer development. Wu et al. reported that a regular intake of AFB_1_ decreased *Sat2* and *LINE-1* promoter methylation in white blood cells from 1140 cancer-free patients of the Cancer Screening Program cohort in Taiwan, suggesting that AFB_1_ plays a major role in hepatocellular carcinoma by inducing global DNA hypomethylation [266].

In another study, primary human hepatocytes were treated with 0.3 µM AFB_1_ for 5 days, followed by 3 days of withdrawal from the carcinogenic exposure [216]. After the initial 5 days exposure to AFB_1_, 5639 genes were differentially methylated, of which 1896 were hypermethylated and 3743 were hypomethylated. Following the 3-day withdrawal, 8131 differentially methylated genes were detected, of which 4397 were hypermethylated and 3734 were hypomethylated. From the initial 5639 differentially methylated genes, 386 remained hypermethylated and 1134 remained hypomethylated after the 3 days of withdrawal. In addition, the persistent hypomethylation and upregulation of 6 hepatocellular carcinoma related genes (*TXNRD1*, *PCNA*, *CCNK*, *DIAPH3*, *RAB27A*, and *HIST1H2BF*) were identified to have an influence at the transcriptome level [216]. These genes are known to induce and promote carcinogenesis by elevating cell growth, invasion, and metastasis.

### 4.5. Zearalenone

Zearalenone is a non-steroidal mycotoxin synthesized by various *Fusarium* species. It is found in breakfast cereals, bread, grain, dried fruits, beer, and wine, thus affecting humans and animals through the food chain [267]. Zearalenone is structurally similar to estrogen and competes with estradiol to bind to estrogen receptors, thereby altering the production of hormones with reproductive toxic effects [268]. Exposure to zearalenone prevents protein and DNA synthesis, and it prompts lipid peroxidation, oxidative damage, apoptosis, endoplasmic reticulum, and mitochondrial stress [269,270]; however, few studies have explored its effect on DNA methylation.

A 40 µM zearalenone treatment for 24 h induced global DNA hypomethylation and was associated with an active apoptosis and a decrease in human bronchial epithelial (BEAS-2B) cell viability [271]. The exposure of MCF7 cells to 50 µmol/L of zearalenone for 24 h decreased cell viability and induced global DNA hypermethylation, as suggested by the significant increase in 5-methylcytosine DNA content. This increase in global DNA methylation was correlated with an increase in *DNMT1* and *MGMT* gene expression levels [217]. The exposure of Caco-2 cells to 40 µM of zearalenone for 24 h has also been correlated to an increase in 5-methylcytosine content and a decrease in cell viability [272]. Similarly, the exposure of HepG2 cells to increasing concentrations of zearalenone (1, 10, and 50 µM; 24 h) induced an increase in global DNA methylation through an increase in DNMT1 expression. Moreover, a decrease in the DNA methylation level of the *PPARγ* promoter correlated with an increase in *PPARγ* expression and a reduction in cell viability [273]. Zearalenone exposure at a concentration of 50 µM for 12 h increased 5-methylcytosine levels in mouse oocytes [274]. In contrast, 20 and 40 µg/kg of zearalenone treatment for 5 days decreased 5-methylcytosine levels in CD1 mouse testis [275].

### 4.6. Deoxynivalenol

Deoxynivalenol (DON), a type B trichothecene, is produced by *Fusarium graminearum* and *Fusarium culmorum*. It is primarily found in wheat, maize, oats, and barley, and its exposure has been linked with vomiting, diarrhea, anorexia, gastrointestinal discomfort, impaired growth, and immune dysfunction [276].

The effect of DON on DNA methylation is limited. Exposure of the Caco-2 cell line to 5–10 µM of DON for 24 h decreased cell viability and increased global DNA methylation, as indicated by the increase in 5-methylcytosine content [272].

Additionally, DON (3 µM; 26–72 h) induced autophagy and apoptosis by increasing global DNA methylation and *DNMT3A* expression in porcine oocytes [277]. In piglet liver, DON (1 and 3 mg/kg; 4 weeks) exposure induced global DNA hypermethylation by dose-dependently increasing the expression of *DNMT1* and *DNMT3B*. Furthermore, the expression of cytochrome P450 enzymes and insulin-like growth factor 1 (IGF1) was elevated by DON due to alterations in promoter methylation [278].

To further understand the epigenetic changes in response to DON exposure, Wang et al. characterized genome-wide DNA methylation and gene expression in the porcine small intestine epithelial (IPEC-J2) cell line. They sourced over 300 differentially methylated regions (DMRs) and profiled their distribution across the genome. The majority of the DMRs (2093 out of 3030) were hypermethylated following DON exposure [218]. Twenty-seven CpG island-associated promoters displaying inverse correlations between DNA methylation and their corresponding gene expressions were identified. These genes were all hypermethylated in their promoter regions, following DON exposure [218]. Wang et al. also recognized DNA hypermethylation in gene bodies of the *ESR1* gene in IPEC-J2 cells exposed to 1000 ng/mL of DON for 48 h.

Notably, DON exposure in the above-mentioned studies all showed DNA hypermethylation, irrespective of the cell or gene type. Depending on the gene, DNA hypermethylation is associated with transcriptional repression, leading to tumor suppressor gene silencing. This suggests an alternative mechanism of DON cytotoxicity that may lead to cancer development and progression. The DON-induced DNA hypermethylation in porcine oocytes may provide a mechanism for ovarian cancer. However, further studies are required to investigate and confirm the role of DON in carcinogenic epigenetic modifications.

### 4.7. T-2 Toxin

T-2 toxin is a trichothecene mycotoxin produced by *Fusarium acuminatum*, *Fusarium poae,* and *Fusarium sporotrichoides* [279]. It is known to contaminate wheat, maize, and barley, resulting in feed rejection, growth retardation, and reproductive and gastrointestinal dysfunction in pigs [280]. In chickens, the liver is also a target of T-2 toxin, and its toxicity has been attributed to oxidative stress [281], DNA damage [282], lipid peroxidation [282], apoptosis, and autophagy [283].

T-2 toxin has been shown to trigger hepatotoxicity in rats via DNA methylation and pro-inflammatory cytokines [219]. In vivo, rat liver exposed to 2 mg/kg body weight of T-2 toxin for 1, 3, and 7 days showed a decrease in the methylation level of the promoter regions of the pro-inflammatory cytokine genes, *IL-11*, *IL-6*, *IL-1α*, and *TNF-α*, thereby increasing the expressions of these inflammatory genes and promoting their transcriptional binding [219]. The assessment of global DNA methylation revealed an increase in the 5-methylcytosine level and an increase in *DNMT1* expression [219]. An increase in the *DNMT3A* gene expression was also observed but only after 3 days of exposure to T-2 toxin [219]. Protein expression of the DNA methyltransferases was increased through days 1, 3, and 7 [219]. The in vitro exposure of normal rat liver cells (BRL) to 10, 20, or 40 nM of T-2 toxin for 8 h or 12 h was associated with a decrease in the methylation of *TNF-α* and *IL-11* gene promoters; however, no changes in the promoter methylation of *IL-6*, *IL-1β*, and *IL-1α* was observed [219]. The expression of *DNMT1* and *DNMT3A* showed varying degrees of elevation over the 8 h and 12 h exposures, while DNMT1 and DNMT3A protein expressions were consistently elevated [219]. The results from this study indicated that T-2 toxin increases the expression of pro-inflammatory cytokines through altered DNMT expression and hypomethylation of their promoters. This study suggests that T-2 toxin, by inducing chronic inflammation, may promote malignant cell transformation [284].

Ras association domain family 4 (RASSF4) is a tumor suppressor protein that is widely expressed in normal tissues; however, loss of expression due to DNA hypermethylation is related to nasopharyngeal carcinoma [285], non-small cell lung cancer [286], colorectal cancer [287], and head and neck cancers [288]. Thus, Liu et al. investigated the role of DNA methylation and *RASSF4* expression in hepatotoxicity caused by T-2 toxin using rat liver (2 mg/kg for 1, 3, and 7 days) and the BRL cell line (10, 20, and 40 nM for 4, 8, and 12 h, respectively). In vivo, the promoter region of the *RASSF4* gene was hypermethylated following 1 day of T-2 toxin intake, and *RASSF4* expression was decreased [289]. This decrease in *RASSF4* may enhance cancer cell growth. After 3 days of exposure to T-2 toxin, the methylation of the *RASSF4* gene promoter was increased, and with this, its expression [289]. This suggested that T-2 toxin might induce cellular apoptosis in rat liver. Exposure to T-2 toxin for 7 days led to promoter DNA hypermethylation and reduced expression of the *RASSF4* gene [289]. In contrast, the in vitro experiment showed that T-2 toxin significantly decreased the promoter methylation of *RASSF4*, thus increasing *RASSF4* expression [289]. These results suggested that T-2 toxin may play a role in carcinogenesis.

## 5. Conclusions

Cancer development is a continuous, progressive event, involving the activation of oncogenes and the inhibition of tumor suppressor genes by alterations of the epigenome. Lifestyle modification is a major step in preventing the disease and improving patient survival. The human diet has been shown to contain micronutrients, bioactive compounds, and mycotoxins that have the potential to alter the epigenome; this can either lead to cancer prevention or development. As discussed in this review, DNA methylation is a common epigenetic mechanism by which micronutrients, bioactive dietary compounds, and mycotoxins affect cancer cell growth, invasive capacities, and metastasis. Although micronutrients and bioactive dietary compounds have shown the potential to inhibit and prevent cancer, the main disadvantage is that the majority of these studies are conducted in vitro. In vitro models lack the complex biological functions and muticellularity observed in vivo and hence, the effects of these compounds in vivo may be different from those seen in vitro. This is further exaggerated by several studies in which changes in DNA methylation were both site and cell specific. Another disadvantage arises from the observation that the cancer-benefiting properties of these compounds occurred at varying concentrations, some of which may prove to be harmful in the long-term. Other challenges include the bioavailability of the exact concentration required to reverse the epigenetic modification and the effect of these micronutrients and bioactive compounds in combination with existing anti-cancer drugs. Likewise, some mycotoxins, which are common in agricultural foods (the same agricultural foods are a source of bioactive compounds), are known carcinogens. Figure 5 depicts an overview of the proposed mechanism for how food-borne mycotoxins affect the DNA methylation profile of the cell, thus leading to the development and progression of cancer.

This review emphasizes the shift away from high-fat, low-fiber, and mycotoxin-contaminated foods that increase inflammation and oxidative stress and may increase cancer risk toward foods that are mycotoxin-free and rich in micronutrients and bioactive compounds that prevent and destroy cancer cells.

## Figures and Tables

**Figure 1 cells-09-02004-f001:**
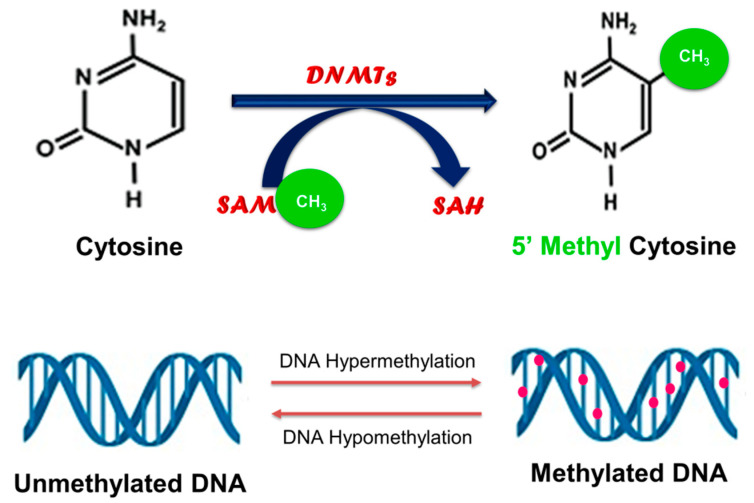
Process of DNA methylation. DNA methyltransferases (DNMT1, DNMT3A, and DNMT3B) catalyze the transfer of a methyl group from the universal methyl donor S-adenosylmethionine (SAM), resulting in the formation of 5-methylcytosine.

**Figure 2 cells-09-02004-f002:**
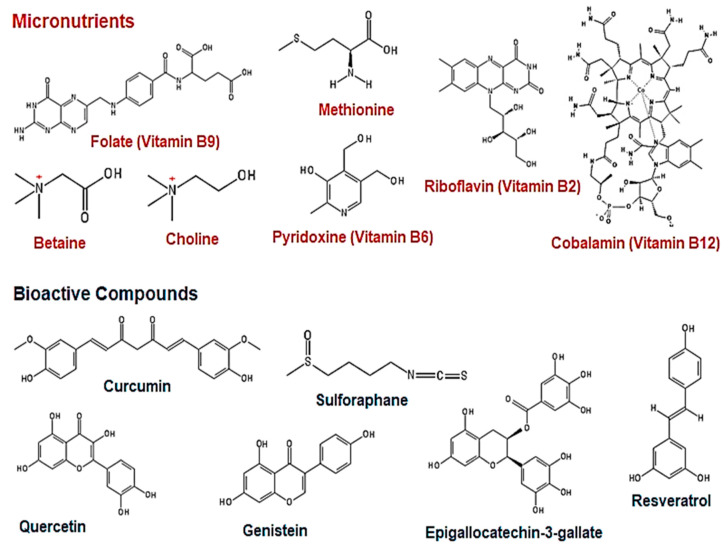
Chemical structures of micronutrients and bioactive dietary compounds. Chemical structures were drawn using PubChem Sketcher Version 2.4.

**Figure 3 cells-09-02004-f003:**
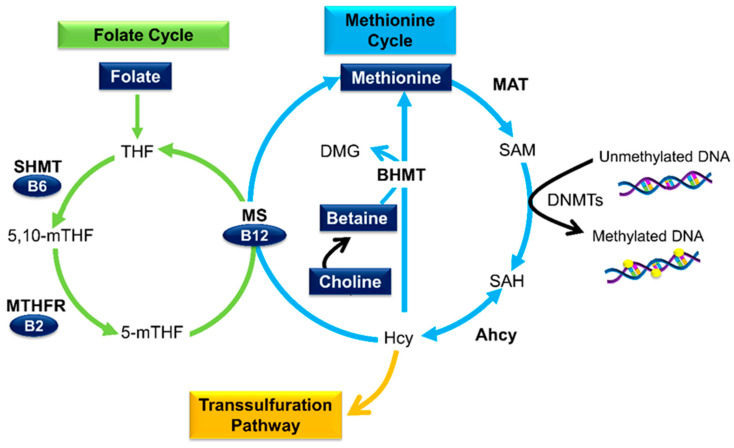
The role of micronutrients in one-carbon (1C) metabolism and DNA methylation. Methionine, the precursor of SAM, is obtained from the diet or through the re-methylation of homocysteine (Hcy). Hcy re-methylation can occur via two pathways: (1) Folate, in the form of tetrahydrofolate (THF), is converted to 5,10-methyltetrahydrofolate (5,10-mTHF) and subsequently to 5-mTHF, which acts as a methyl donor for Hcy re-methylation. The reactions are catalyzed by serine hydroxymethyl transferase (SHMT), methylene tetrahydrofolate reductase (MTHFR), methionine synthase (MS), and their respective cofactors (vitamins B6, B2, and B12); (2) betaine homocysteine methyl transferase (BHMT) catalyzes the re-methylation of Hcy by using betaine (dietary or via choline oxidation) as a methyl donor resulting in methionine and dimethylglycine (DMG). Methionine (endogenous or dietary) is converted to SAM via methionine adenosyltransferase (MAT). The transfer of the methyl group from SAM to DNA results in SAH and Hcy. Hcy also bifurcates to the transsulfuration pathway.

**Figure 4 cells-09-02004-f004:**
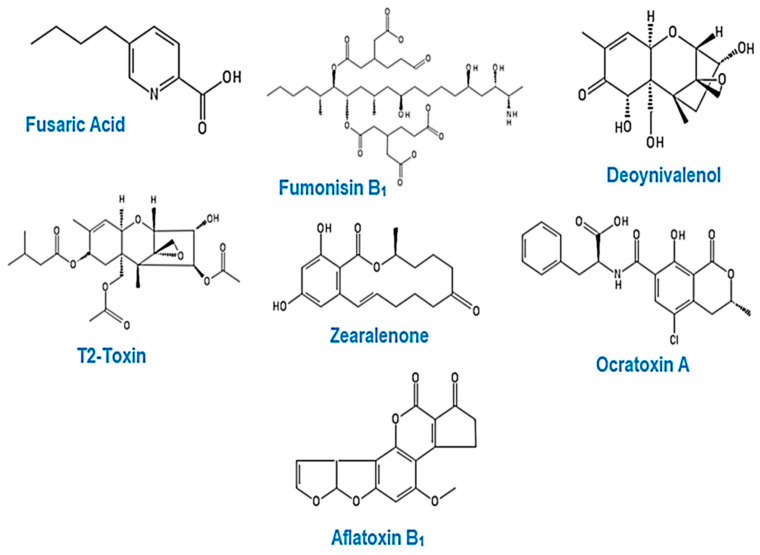
Chemical structures of common food-borne mycotoxins. Chemical structures were drawn using PubChem Sketcher Version 2.4.

**Figure 5 cells-09-02004-f005:**
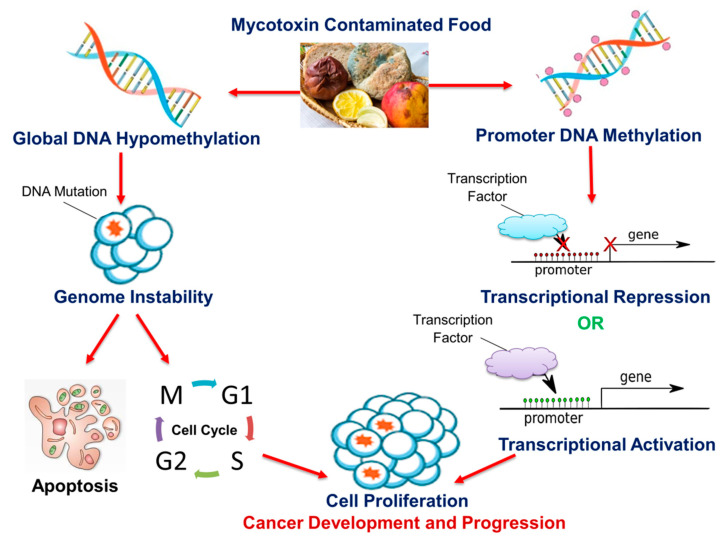
The effect of mycotoxins on DNA methylation and cancer. Mycotoxins can alter global DNA methylation and/or promoter DNA methylation in various cells both in vitro and in vivo. Global DNA hypomethylation leads to genome instability and increases the frequency of DNA mutations. Mutated cells are destroyed via apoptosis or evade cell cycle regulatory checkpoints and proliferate, leading to cancer development and progression. Promoter DNA methylation contributes to carcinogenesis via the transcriptional repression of tumor suppressor genes and/or transcriptional activation of oncogenes.

**Table 1 cells-09-02004-t001:** The effect of folic acid on global and gene promoter DNA methylation in various human cancer cell lines.

Cancer Type and Model	Concentration of Folic Acid	Duration of Treatment	Effect on DNA Methylation and Cellular Outcome	Reference
Breast cancer: MCF-7 and MDA-MB-231 cells	4–8 mg/L	4 days	MCF-7 and MDA-MB-231 cells: ↑ *DNMT1* expression; ↑ Promoter methylation of *PTEN*, *APC*, and *RARβ2*; ↑ caspase-3-dependent apoptosis	[33]
Colorectal cancer: HCT116, LS174T, and SW480 cells	4–16 mg/L	7 days	HCT116 cells: ↑ DNMT1 expression; ↓ DNMT3A and DNMT3B expression; ↓ Global DNA methylation; ↑ Cell proliferation; ↑ Colonosphere formation LS174T cells: ↑ DNMT1 expression; DNMT3A not expressed; No change in DNMT3B expression; ↓ Global DNA methylation; ↑ Cell proliferation; ↑ Colonosphere formation SW480 cells: ↓ DNMT1, DNMT3A, and DNMT3B expression; No change in global DNA methylation; ↑ Cell proliferation; ↑ Colonosphere formation	[34]
Colon cancer: HCT116 and Caco-2 cells	0–2.3 µM	20 days	Folic acid deficient (0 µM) HCT116 cells: ↓ DNMT1 and DNMT3A expression; No change in DNMT activity; No change in global DNA methylation; ↑ *ER* promoter methylation; No change in *ER* expression; ↓ Cell growth Folic acid deficient (0 µM) Caco-2 cells: ↓ DNMT1 and DNMT3A expression; ↓ DNMT activity; No change in global DNA methylation; ↑ *ER* promoter methylation; No change in *ER* expression; ↓ Cell growth	[46]
Colon cancer: SW620 cells	0–3 µmol/L	14 days	Folic acid deficiency (0 µmol/L): ↓ Global DNA methylation; ↓ *p53* gene-specific DNA methylation. In both cases, the effects of folic acid depletion were reversed by folic acid (3 µmol/L) supplementation	[47]
Colon cancer: HCT116 and SW480 cells	Commercial folate-deficient RPMI 1640 medium	HCT116 cells: 24–48 h SW480 cells: 24–72 h	HCT116 and SW480 cells: ↓ *Shh* gene promoter methylation; ↑ Shh gene and protein expression; ↑ Activation of Shh signalling; ↑ Migration and invasiveness	[48,49]
Colon cancer: Caco-2 cells	20 µM	48 h	↑ Promoter methylation of *ESR1*, *p15^INK4b^*, and *p16^INK4a^*; ↑ Cell proliferation	[50]
Nasopharyngeal cancer: KB cells	2–10 nM	-	↑ Promoter methylation of *H-cadherin*; ↓ *H-cadherin* expression; Promotes malignant phenotype	[51]

↑: Increase; ↓: Decrease; PTEN: Phosphatase and tensin homolog; APC: Adenomatous polyposis coli; RARβ2: Retinoic acid receptor beta 2; ER: Estrogen receptor; p53/p15^INK4b^/p16^INK4a^: Tumor suppressor proteins; Shh: Sonic hedgehog; ESR1: Estrogen receptor 1.

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
