# Peer review of "The Impact of Natural Dietary Compounds and Food-Borne Mycotoxins on DNA Methylation and Cancer"

_cells, 2020, doi:10.3390/cells9092004_

Round 1

Reviewer 1 Report

The authors made a state of the art on the different component identified in food that can affect the DNA methylation level in human, and with this perturb the gene expression and promote tumor formation.

In general the manuscript is well structured, however, the authors need to be more precise in the data cited (cohort information on age, origin/ethnicity, sex) and the cells on which the experiments were run (human?, in vitro cell lines / primary cells from tumors; animal model).

They need to support their declaration with references in places. 

The ref format need to be harmonized.

Please find in the attached file my details comments through the manuscript.

Author Response

Reviewer 1

In general the manuscript is well structured, however, the authors need to be more precise in the data cited (cohort information on age, origin/ethnicity, sex) and the cells on which the experiments were run (human in vitro cell lines / primary cells from tumors/ animal model)

The above information has been added to the manuscript as follows:

Page 11 lines 216-217: “In adults, mean age of 57.8 years and belonging to the ethnic groups of White, African American, or Hispanic, who …”

Page 11 lines 222-223: “…colorectal adenomas in adults aged between 21-80 years and belonging to the ethnic groups of White, Black, Hispanic, or Asian [57].

Page 11 line 226: “…carcinomas in Caucasians aged 65 years and over [58].”

Pages 11-12 lines 227-228: “…lung cancer in Norwegians with a mean age of 62.3 years [59].”

Page 12 lines 231-232: “…patients (aged 49-82 years) with colorectal…”

Page 12 lines 234-235: “…patients (mean age of 62 years) with colonic…”

Page 12 line 236: “…patients (mean age of 63.9 years) with…”

Page 12 lines 240-241: “…women (aged between 30-91 years and of Caucasian, African American, Hispanic, or Asian origin)…”

Page 12 line 248: “…in lung cancer patients aged 35-70 years [65].”

Page 17 line 381: “…several human in vitro…”

Page 17 line 383: “…several different human tumor cell lines…”

Page 17 line 398: “…in mice and human prostate cancer cell lines, respectively.”

Page 18 line 403: “…decreased HSC-T6 and primary human stellate cell viability…”

Page 18 line 408: “…leukemia and colorectal cancer cell lines…”

Page 18 lines 415-416: “…cell growth inhibition in various human cancer cell lines and animal models [136-138].”

Page 18 line 417: “…G2/M phase in human breast and laryngeal cancer cells…”

Page 18 line 418: “In human bladder cancer cell lines…”

Page 18 line 422: “…hypomethylation in human cervical cancer (HELA) cells…”

Page 20 lines 473-474: “…hypomethylation in human breast, prostate, and cervical cancer cell lines [13,163-166].”

Page 20 line 485: “In human cervical cancer (HELA) cell lines…”

Page 21 lines 491-492: “…observed in MCF7 and MDA-MB-231 breast cancer cell lines where…”

Page 21 lines 501-502: “…human MCF7 and MDA-MB-231 breast cancer cell lines…”

Page 21 lines 502-503: “…observed in LnCap and DU145 prostate cancer cell lines,…”

Page 21 line 504: “In nasopharyngeal primary carcinoma cells…”

Page 21 line 512: “…apoptosis in MCF7 and MDA-MB-231 breast cancer cell lines…”

Page 21 line 514: “…inhibition against mouse melanoma B16F10 cell growth…”

Page 22 lines 515-516: “…nasopharyngeal primary carcinoma cells…”

Page 22 lines 521-522: “In breast cancer (MCF‐7 and MDA‐MB‐231) cell lines…”

Page 22 line 529: “…vitro (MCF7, UACC3199, and HCC38 cell lines) and in vivo (BRCA1 F22/24 mice)…”

Page 22 line 532: “…cancer cell lines…”

Page 22 lines 537-538: “…BTG3 in human prostate (LnCap and PC3) cancer [188], renal (A498 and ACHN) carcinoma [189], and leukemia (MOLT17, MOLT4, and Jurkat) [190] cell lines…”

Page 22 lines 541-542: “In the HT29 colon cancer cell line…”

Pages 22-23 lines 543-544: “Contrastingly, in Sprague Dawley rats, genistein…”

Page 23 line 545: The following was added “…apoptosis of colon cancer cells through…”

Page 23 line 550: “…carcinoma (KYSE 510) cell line by…”

Page 23 line 553: “…KYSE 150 cell line and prostate cancer (LNCaP and PC3) cell lines…”

Page 23 line 555: “…inhibited the growth of the KYSE 510 cell line…”

Page 23 line 571: “…of the A431 skin cancer cell line with…”

Page 24 line 574-575: “In the cervical cancer (HELA) cell line, EGCG…”

Page 24 line 579: “In human lung cancer (A549 and H460) cell lines,…”

Page 24 line 586: “…esophageal cancer (ECa109) cell line through…”

Page 24 line 587: “In breast cancer (MCF7 and MDA-MB-231) cell lines, EGCG…”

Page 24 line 592: “Similarly in HSC3 and SCC9 oral carcinoma cell lines…”

Page 24 line 595: “…MCF7 and MDA-MB-231 breast cancer cell lines…”

Page 25 lines 604-605: “…prostate cancer (PC3 and DU145) cell lines with…”

Page 25 lines 619-620: “In a randomized clinical trial on 10 male Norwegian patients aged between 55-69 years who received…”

Page 25 line 626: “…34 healthy premenopausal women (aged between 19-54 years) fed…”

Page 26 lines 630-631: “…39 women (median age of 59.5 years in the 5 mg group versus a median age of 54 years in the 50 mg group) with…”

Page 31 line 771: “…OTA (25 µM) in the HepG2 cell line was…”

Page 34 lines 850-851: “…porcine small intestine epithelial (IPEC-J2) cell line.”

They need to support their declaration with references in places

The following references have been added to the manuscript:

Page 5 lines 123-127: “Accumulating evidence indicates that the human diet is a source of micronutrients (folate, B vitamins, betaine, choline, and methionine; Figure 2) and bioactive compounds (curcumin, epigallocatechin-3-gallate, genistein, quercetin, resveratrol, and sulforaphane; Figure 2) that act as both chemopreventative and chemotherapeutic agents by modulating the epigenome [13-16,18].”

Page 5 lines 127-130: “This can occur through alterations in DNA methylation patterns and is usually the consequence of a direct interaction with the enzymes responsible for establishing DNA methylation marks or by acting as methyl donors and co-factors for DNA methylation reactions [31-34].”

Page 26 lines 644-645: “These mycotoxins include fusaric acid, fumonisin B1, deoxynivalenol, T-2 toxin, zearalenone, ochratoxin A, and aflatoxin B1 (Figure 4) [212]. Omotayo, O.P.; Omotayo, A.O.; Mwanza, M.; Babalola, O.O. Prevalence of mycotoxins and their consequences on human health. Toxicological Research 2019, 35, 1-7.

The reference format needs to be harmonized

All references in text have been formatted as per the journal guidelines:

Page 11 line 221: The following was deleted “…however, Cole et al. (2007) showed that 1 mg/day…”

Page 12 line 230: The following was deleted “Cravo et al. (1994) showed that…”

Page 12 line 235: The following was deleted “Pufulete et al. (2005) showed…”

Page 15 line 341: The following was deleted “…study by Vineis et al. (2011), the…”

Page 16 line 347: The following was deleted “…Bassett et al. (2012) and Fanidi et al. (2018)…”

Page 16 line 362: The following was deleted “…Chaturvedi et al. (2018))…”

Page 28 line 671: The following was deleted “…toxicity, Ghazi et al. (2019) investigated…”

Page 29 line 701: The following was deleted “A study by Mobio et al. (2000) indicated…”

Page 29 line 704: The following was deleted “…Chuturgoon et al. (2014) investigated…”

Page 29 line 710: The following was deleted “…Chuturgoon et al. (2014) implicates…”

Page 29 line 712: The following was deleted “Demirel et al. (2015) determined…”

Page 30 line 741: The following was deleted “…Giromini et al. (2016)…”

Page 31 line 746: The following was deleted “…Ozden et al. (2015)…”

Page 31 line 755: The following was deleted “…Li et al. (2015) investigated…”

Page 31 line 765: The following was deleted “Li et al. (2015) also…”

Page 31 line 770: The following was deleted “…Zheng et al. (2013)…”

Page 32 line 786-787: The following was deleted “…Zhang et al. (2002) investigated…”

Page 32 lines 794-795: The following was deleted “Wu et al. (2013) reported…”

Page 34 lines 849-850: The following was deleted “…Wang et al. (2019) characterized…”

Page 34 lines 856-857: The following was deleted “Wang et al. (2019) also…”

Page 36 line 904: The following was deleted “Liu et al. (2019) thus…”

Detailed comments mentioned in the PDF copy of the manuscript

Page 2 line 78 (PDF): “…exclusively at CpG dinucleotides…” This is true in eukaryotes, on bacterial and plant DNA, methylation can also occur at an adenine site.

The following information was amended “In eukaryotes, DNA methylation occurs almost exclusively at CpG dinucleotides…” – Page 4 line 93.

Page 6 line 137 and Page 7 line 140: The numbering of the Figure has been changed to Figure 3 to accommodate the addition of Figure 2.

Page 7 line 196 (PDF): “However, the intake of either folate or vitamin B12 was above the recommended dietary intake.” In which study?

The following was added “However, in both studies, the intake of either folate or vitamin B12 was above the recommended dietary intake.” – Page 12 lines 228-229.

Page 12 line 252: The following was added “…such as age, gender, family history, ethnic group, and lifestyle factors…”

Page 16 line 344-345: A repeated citation was removed “…current smokers [100].” and correctly replaced “…current smokers [65].”

Page 10 line 329 (PDF): “in vivo” Mention if this was shown in human or animal model.

Page 17 lines 379-380: The following was deleted “…both in vitro and in vivo.”

Page 17 line 379: The following was added “…properties both in human cancer cell lines and animal models.”

Page 21 line 492: The following was deleted “…breast cancer (MCF7 and MDA-MB-231) cells…”

Page 23 line 544: The following was deleted “…in vivo colon cancer cells, genistein…”

Page 23 line 550: The following was deleted “…carcinoma (KYSE 510 cells)…”

Page 23 line 555: The following was deleted “…expression of RARβ and, p16INK4a, and MGMT and inhibited the cell growth…”

Page 23 line 555: The following was added “…expression of RARβ and p16INK4a and inhibited the growth…”

Page 25 line 608: The following was added “Exposure of the MDA-MB-231 cell line to a mixture of…”

Page 25 lines 608-609: The following was deleted “A combination of EGCG (20.0 µg/ml) and SFN (5.0 µM) in MDA-MB-231 cells for 72 h…”

Page 25 lines 611-612: The following was deleted “Similarly, treatment (48 h) of MCF7 cells with a combination of genistein…”

Page 25 lines 611-612: The following was added “Similarly, a 48 h treatment of the MCF7 cell line with a mixture of genistein…”

Page 25 line 624: The following was deleted “…developmental processes and cell…”

Page 26 line 637: The following reference was deleted “…threatening food security [213].”  and replaced by “…threatening food security [212].”

The following reference was deleted from the reference list “213. Huang, D.; Cui, L.; Sajid, A.; Zainab, F.; Wu, Q.; Wang, X.; Yuan, Z. The epigenetic mechanisms in Fusarium mycotoxins induced toxicities. Food and Chemical Toxicology 2019, 123, 595-601, doi:https://doi.org/10.1016/j.fct.2018.10.059.”

Page 55 lines 1697-1699: The following reference was added to the reference list “212. Omotayo, O.P.; Omotayo, A.O.; Mwanza, M.; Babalola, O.O. Prevalence of mycotoxins and their consequences on human health. Toxicological Research 2019, 35, 1-7.”

Page 14 line 555 (PDF): “several environmental factors” Be more precise and add references

Page 14 line 555-556 (PDF): “several environmental factors such as warm humid conditions; and are common in areas where there are poor food handling and storage methods.”  Not really, they are present all over the world. I suppose that what you mean is that there are common in the food of areas with poor regulation on the tolerable level of mycotoxin content.

Page 26 lines 639-642: The above 2 queries were addressed by addition of the following information “…several environmental factors such as warm humid conditions; and constant exposure to mycotoxins is common in areas where there are no regulations that protect the food intake of the populace [212]. Mycotoxin exposure is also common in areas with poor food handling and storage methods [212].”

Page 26 lines 646-647: The following was deleted “…epigenetic modifications have become the focus and trend of toxicity studies [213], and these changes in epigenetic modifications especially, DNA…”

Page 26 lines 646-649: The following was added “…epigenetic modifications, in particular in the DNA methylation profile of the cell, can lead to neoplastic transformation and cancer development (Figure 5).

Page 28 line 673: The following was added “…DNA content which is indicative of global DNA hypomethylation in the HepG2 cell line.”

Page 28 line 677: The following was added “….promoter hypermethylation of the DNMTs.”

Page 28 line 678: The following was added “…to occur due to MBD2 promoter hypomethylation…”

Page 28 lines 679-680: The following was added “…cytotoxicity in the HepG2 cell line by altering their DNA methylation profile.”

Page 15 line 582 (PDF): “Fumonisin B1 (FB1) is produced by Fusarium verticilliodes” This mycotoxin is produced by several members of Fusarium fujikuroi species complex (FFSC), including F. verticilliodes and even by other fungal species such as Aspergillus nigri that also produces fumonisins in the crop plants of peanut, maize, and grape.

Page 28 line 682: This statement was amended as follows “Fumonisin B1 (FB1) is produced by several Fusarium species…”

Page 15 line 583 (PDF): “maize and maize products” The Fusarium species is not the same all over the world. By the way, Fumonisin B1 is not only present in maize and maize products, but more generally in crops (Kamle et al. 2019 10.3390/toxins11060328).

Page 28 line 683: This statement was amended to include agricultural crops “…fungal contaminant of maize and other agricultural crops…”

Page 28 line 682: The following was deleted “…Fusarium verticilliodes…”

Page 28 line 683: The following was deleted “… of maize and maize products…”

Page 28 lines 684-685: The following was added “It has been classified as a type 2B carcinogen by the International Agency for Research on Cancer (IARC) [227].”

Page 57 lines 1743-1745: The following reference was added to the reference list “227. Cancer, I.A.f.R.o. Some traditional herbal medicines, some mycotoxins, naphthalene and styrene. Working group on the evaluation of carcinogenic risks to humans. IARC Monogr Eval. Carcinog. Risks Hum 2002, 82, 1-556.”

Page 29 line 689: The following was added “FB1 has been proved to induce tumor proliferation…”

Page 29 line 690: The following was deleted “….carcinogenic in the liver…”

Page 15 lines 588-590 (PDF): “Consumption of corn derived foods, contaminated by FB1 has also been correlated with a high incidence of oesophageal cancer in South Africa and China [230,231].” All food is contaminated by mycotoxins. Mention which level of Fumonisin B1 in food was associated with the oesophageal cancer incidence in general population (e.g. Mozambique population).

Page 29 lines 690-696: The following was added “FB1 contamination in corn (223.6 μg/g) and rice (21.6 μg/g) was associated with an increased risk of esophageal cancer in the Golestan Province of Northeastern Iran [230]. Consumption of corn contaminated by FB1 at concentrations of 8.67 µg/kg body weight per day in the Transkei region of South Africa and 18-155 ppm in the Cixian and Linxian counties of the People's Republic of China has also been correlated with a high incidence of esophageal cancer [231,232].”

Page 29 line 692: The following was deleted “…corn derived foods, contaminated…”

Page 29 line 695-696: The following was deleted “…cancer in South Africa and China…”

Page 15 line 593 (PDF): “deficiency of folate by FB1” Please modified, the meaning is not strength forward.

Page 29 line 699: “that folate deficiency due to FB1 leads…”

Page 29 line 699: The following was deleted “…due to of folate by FB1...”

Page 29 lines 701-702: The following was deleted “…FB1 (9-18 µM; for 24 h) induced…”

Page 29 lines 701-702: The following was added “…that a concentration of 9-18 µM of FB1 for 24 h…”

Page 29 lines 702-703: The following was added “…in the rat C6 glioma cell line; however…”

Page 29 lines 705-706: The following was added “…in the HepG2 cell line. They documented that FB1 (200 µM) induces global DNA hypomethylation in the HepG2 cell line via…”

Page 29 lines 712-713: The following was deleted “…from (1, 5, 10, 25, and 50 µM; 24 h) on cell viability and CpG island methylation…”

Page 29 lines 712-714: The following was added “…effect of increasing concentrations of FB1 from 1-50 µM for 24 h on cell viability and on the CpG methylation levels of tumor suppressor genes and oncogenes in liver and kidney (Clone 9 and NRK-52E) cell lines.

Page 30 lines 716-717: The following was added “…molecule, E-cadherin, or the cyclin-dependent kinase inhibitor, p15, in both Clone 9 and NRK-52E cell lines. The cyclin D-dependent kinase inhibitor, p16,…”

Page 30 lines 718-719: The following was added “In Clone 9 cells, the level of CpG…”

Page 30 lines 720-721: The following was deleted “…concentrations 10 and 50 µM. The promoter methylation of c-myc was un-methylated for…”

Page 30 lines 720-721: The following was added “…concentrations of 10 and 50 µM. The methylation of the c-myc promoter was decreased for…”

Page 30 lines 723-724: The following was added “…concentrations of 10, 25, and 50 µM and in NRK-52E cells at concentrations of 10 and 50 µM. Despite the changes in the DNA methylation profile…”

Page 30 line 731: The following punctuation was added “…small intestine,…”

Page 30 lines 733-734: The following was added “OTA has been shown to induce…”

Page 30 line 734: The following was deleted “OTA thus induces kidney…”

Page 30 line 735: The following was added “…carcinogen by the IARC [250].”

Page 58 lines 1815-1817: The following reference was added to the reference list “250. Humans, I.W.G.o.t.E.o.C.R.t.; Cancer, I.A.f.R.o.; Organization, W.H. Some naturally occurring substances: food items and constituents, heterocyclic aromatic amines and mycotoxins; World Health Organization: 1993; Vol. 56.”

Page 30 line 736: The following was deleted “…proposed; however, the mechanism…”

Page 30 line 736: The following was added “…proposed; as the mechanism…”

Page 30 lines 738-740: The following was deleted “DNA methylation has been a topic of interest in OTA toxicity studies and has become a marker…”

Page 30 lines 738-739: The following was added “The effect of OTA on DNA methylation levels in human cells has become a topic of interest. Consequently, the level of DNA methylation has become a marker…”

Page 30 lines 740-742: The following was added “…OTA and DNA methylation, Giromini et al. estimated the modifications in the DNA methylation level of BME-UV1 and MDCK cell lines by concentration of OTA of…”

Page 30 lines 742-744: The following was deleted “…of evaluated the effects of OTA addition in bovine mammary epithelial (BME-UV1) cells and Madin-Darby canine kidney (MDCK) cells by assessing the global DNA methylation status.”

Page 31 line 745: The following was deleted “Following treatment with OTA (1.25 µg/ml; 24 h)…”

Page 31 line 745: The following was added “…1.25 µg/ml for 24 h, both BME-UV1 and MDCK cell lines…”

Page 31 line 748: The following was deleted “…OTA (0, 21, 70, or 210 μg/kg) treatment.”

Page 31 line 748: The following was added “…of an OTA intake of 21, 70, or 210 μg/kg.”

Page 31 lines 749-750: The following was added “…analysis of DNA methylation at 90 days showed that OTA induced differential DNA methylation levels changes in the promoters of genes…”

Page 31 line 750: The following was deleted “…levels changes in…”

Page 31 line 752: The following was deleted “…role for OTA…”

Page 31 line 752: The following was added “…role of OTA…”

Page 31 line 753: The following was added “…Dlg2 gene promoters, and…”

Page 31 line 754: The following was added “…gene promoters…”

Page 31 lines 755-756: The following was deleted “…low dosage (70 µg/kg) and high dosage…”

Page 31 line 756-757: The following was added “…OTA intake on global DNA methylation in rat kidneys, after 4, 13, and 26 weeks of consumption.”

Page 31 lines 757-758: The following was deleted “…weeks, as a mechanism of OTA-mediated nephrotoxicity. After 4 weeks, high dose OTA treatment reduced global…”

Page 31 lines 758-759: The following was added “…methylation was reduced in rat kidneys exposed to high dose OTA…”

Page 31 lines 760-761: The following was added “…weeks, the kidneys of both low and high dose OTA treated rats showed a significant increase in global…”

Page 31 line 761: The following was deleted “…weeks, both low and high dose OTA treatment significantly increased global…”

Page 31 line 767: The following was added “….promoters after…”

Page 31 line 768: The following was deleted “…promoter hypermethylation…”

Page 31 line 771: The following was added “…in the HepG2 cell line was…”

Page 31 line 772: The following was added “…decrease in the DNA 5-methylcytosine…”

Page 32 line 785: The following was deleted “…AFB1 exposure…”

Page 32 line 785: The following was added “…AFB1 intake…”

Page 32 line 786: The following was added “…AFB1 concentration and the incidence of hepatocellular…”

Page 32 line 787: The following was deleted “…the promoter methylation…”

Page 32 line 787: The following was added “…the DNA methylation of RASSF1A and p16INK4a promoters…”

Page 32 lines 788-789: The following was deleted “…Taiwan, followed by the correlation of RASSF1A and p16INK4a methylation statuses with…”

Page 32 lines 788-789: The following was added “…Taiwan. They found a correlation between the RASSF1A and p16INK4a methylation statuses and the…” 

Page 32 lines 790-792: The following was deleted “…levels. The results showed hypermethylation of the RASSF1A and p16INK4a promoters in individuals with higher AFB1-DNA adduct levels in hepatocellular carcinoma tumour tissues, but only the RASSF1A data was statistically significant [261].”

Page 32 lines 793-794: The following was added “…DNA methylation status and the expression…”

Page 32 line 795: The following was deleted “…that dietary exposure to AFB1…”

Page 32 lines 795-796: The following was added “…that regular intake of AFB1 decreased Sat2 and LINE-1 promoter methylation…”

Page 17 lines 672-673 (PDF): “white blood cells” In which population?

Page 32 lines 796-797: The following was added “…white blood cells from 1,140 cancer free patients of the Cancer Screening Program cohort in Taiwan,…”

Page 33 lines 817-818: The following was deleted “…however, minimal studies have determined its…”

Page 33 lines 817-818: The following was added “…however, few studies have explored its…”

Page 33 lines 819-820: The following was deleted “Zearalenone (40 µM) treatment for 24 h induced global DNA hypomethylation and was associated with active apoptosis and decreased in human…”

Page 33 lines 819-820: The following was added “A 40 µM zearalenone treatment for 24 h induced global DNA hypomethylation and was associated with an active apoptosis and a decreased in human…”

Page 33 lines 821-824: The following was deleted “In MCF7 cells, zearalenone (50 µmol/L; 24 h) decreased cell viability by inducing global DNA hypermethylation, as indicated by the significant increase in 5-methylcytosine DNA content. The…”

Page 33 lines 821-824: The following was added “The exposure of MCF7 cells to 50 µmol/L of zearalenone for 24 h decreased cell viability and induced global DNA hypermethylation, as suggested by the significant increase in 5-methylcytosine DNA content. This…”

Page 33 lines 825-827: The following was deleted “In Caco-2 cells, zearalenone (40 µM; 24 h) increased 5-methylcytosine content and decreased cell…”

Page 33 lines 825-827: The following was added “The exposure of Caco-2 cells to 40 µM of zearalenone for 24 h has also been correlated to an increase in 5-methylcytosine content and a decrease in cell…”

Page 33 lines 827-829: The following was deleted “Similarly, in HepG2 cells, zearalenone (1, 10, and 50 µM; 24 h) increased global DNA methylation by increasing DNMT1…”

Page 33 lines 827-829: The following was added “Similarly, exposure of HepG2 cells to increasing concentrations of zearalenone (1, 10, and 50 µM; 24 h) induced an increase in global DNA methylation through an increase in DNMT1…”

Page 33 line 830: The following was added “…decrease in the DNA methylation level of the PPARɣ…”

Page 33 line 830: The following was deleted “…promoter methylation correlated…”

Pages 33-34 lines 831-833: The following was deleted “Zearalenone (50 µM; 12 h) exposure….. in mouse oocytes [272]. In contrast, zearalenone (20 and 40 µg/kg)…”

Page 34 lines 832-833: The following was added “…exposure at a concentration of 50 µM for 12 h increased 5-methylcytosine levels in mouse oocytes [272]. In contrast, 20 and 40 µg/kg of zearalenone…”

Page 34 lines 840-841: The following was deleted “In Caco-2 cells, DON (5-10 µM; 24 h) decreased cell viability via increased…”

Page 34 lines 840-841: The following was added “Exposure of the Caco-2 cell line to 5-10 µM of DON for 24 h decreased cell viability and increased…”

Page 34 line 850: The following was added “…in the porcine…”

Page 34 line 851: The following was deleted “…epithelial cells (IPEC-J2)…”

Page 34 lines 852-853: The following was added “The majority…”

Page 34 lines 858: The following was added “…to 1,000 ng/ml of DON for 48 h.”

Page 34 lines 858-859: The following was deleted “ESR1 plays a key role in the endocrine and reproductive systems and drug response in breast cancer.”

Page 35 line 861: The following was deleted “In general,…”

Page 35 line 861: The following was added “Depending on the gene,…”

Page 35 line 865: The following was deleted “…and mouse oocytes…”

Page 35 line 871: The spelling of “Fusarium” was amended

Page 35 lines 873-874: The following was deleted “…feed rejection and growth retardation [279], reproductive and gastrointestinal dysfunction [280], and vomiting [281]. The…”

The following references were deleted from the reference list “279. Wahibah, N.N.; Tsutsui, T.; Tamaoki, D.; Sato, K.; Nishiuchi, T. Expression of barley Glutathione S-1566 Transferase13 gene reduces accumulation of reactive oxygen species by trichothecenes and paraquat in 1567 Arabidopsis plants. Plant Biotechnology 2018, 35, 71-79.” And “281. Fatima, Z.; Guo, P.; Huang, D.; Lu, Q.; Wu, Q.; Dai, M.; Cheng, G.; Peng, D.; Tao, Y.; Ayub, M. The critical 1572 role of p16/Rb pathway in the inhibition of GH3 cell cycle induced by T-2 toxin. Toxicology 2018, 400, 28-39.”

Page 35 lines 873-874: The following was added “…feed rejection, growth retardation, and reproductive and gastrointestinal dysfunction in pigs [280]. In chickens, the…”

Page 35 line 877: The following was deleted “A study by Liu et al. (2019) revealed that T-2 toxin may…”

Page 35 line 877: The following was added “T-2 toxin has been shown to trigger…”

Page 35 lines 879-880: The following was deleted “showed decreased methylation in the…”

Page 35 lines 879-880: The following was added “showed a decrease in the methylation level of the…”

Page 35 lines 882-884: The following was deleted “Assessment of global DNA methylation revealed increased 5-methylcytosine levels and increased DNMT1 expression [285]. DNMT3A expression was also increased…”

Page 35 lines 882-884: The following was added “promoting their transcriptional binding [285]. The assessment of global DNA methylation revealed an increase in the 5-methylcytosine level and an increased in DNMT1 expression [285]. An increase in the DNMT3A gene expression was also observed…”

Page 35 lines 886-888: The following was deleted “In vitro normal rat liver cells (BRL) were exposed to 10, 20 and 40 nM of T-2 toxin for 8 hours and 12 hours [285]. In BRL cells, T-2 toxin decreased the…”

Page 35 lines 886-888: The following was added “The in vitro exposure of normal rat liver cells (BRL) to 10, 20 or 40 nM of T-2 toxin for 8 h or 12 h was associated with a decrease in the…”

Page 36 line 890: The following was added “The expression of DNMT1…”

Page 36 line 891: The following was deleted “…8- and 12 -hour…”

Page 36 lines 894-897: The following was deleted “…cytokines, which is mediated by promoter hypomethylation, and regulates DNMTs thus, disrupting DNA methylation levels of inflammatory genes. Considering that cancer is an inflammatory disease, this study suggests that T-2 toxin, through DNA methylation and chronic…”

Page 36 lines 894-897: The following was added “through altered DNMT expression and hypomethylation of their promoters.  This study suggests that T-2 toxin, by inducing chronic…”

Page 36 lines 899-901: The following was deleted “Many studies have been conducted on T-2 toxin-induced apoptosis, however, the potential role of Ras association domain family 4 (RASSF4) and DNA methylation has not been revealed. RASSF4 is…”

Page 36 lines 907-915: The following was deleted “…following exposure to T-2 toxin for 1 day, and RASSF4 expression was decreased [291], which may enhance cell growth. After 3 days of exposure to T-2 toxin, the RASSF4 gene promoter was hypomethylated, increasing RASSF4 expression [291], and indicating a mechanism for T-2 toxin–induced apoptosis. After T-2 toxin treatment for 7 days, the RASSF4 promoter was once again hypermethylated followed by reduced RASSF4 expression [291]. In contrast, the in vitro aspect showed that…”

Page 36 lines 907-915: The following was added “…following 1 day of T-2 toxin intake, and RASSF4 expression was decreased [291]. This decrease in RASSF4 may enhance cancer cell growth. After 3 days of exposure to T-2 toxin, the methylation of the RASSF4 gene promoter was increased and with this its expression [291]. This suggested that T-2 toxin might induce cellular apoptosis in rat liver. Exposure to T-2 toxin for 7 days led to promoter DNA hypermethylation and reduced expression of the RASSF4 gene [291]. In contrast, the in vitro experiment showed…”

Page 37 lines 917-918: The following was deleted “…T-2 toxin induced alterations in DNA methylation of the RASSF4 gene may…”

Reviewer 2 Report

I have no comments and would like to accept the manuscript in its current form

Author Response

Thank you.

Reviewer 3 Report

It is a well-conducted and written review. Suggestions - integrate a scheme where show the interaction of the metabolic pathways and food borne Mycotoxins on DNA methylation and cancer. Either in parts or in a single scheme.

Author Response

Reviewer 3

Integrate a scheme where show the interaction of the metabolic pathways and food borne mycotoxins on DNA methylation and cancer. Either in parts or in a single scheme.

The following diagram has been included in the manuscript:

(Please see uploaded Rebuttal document for the diagram)

Page 26 lines 645-649: “Recent evidence suggests that mycotoxin-induced epigenetic modifications, in particular in the DNA methylation profile of the cell, can lead to neoplastic transformation and cancer development (Figure 5).”

Pages 27-28 lines 653-660:

Figure 5: The effect of mycotoxins on DNA methylation and cancer. Mycotoxins can alter global DNA methylation and/or promoter DNA methylation in various cells both in vitro and in vivo. Global DNA hypomethylation leads to genome instability and increases the frequency of DNA mutations. Mutated cells are destroyed via apoptosis or evade cell cycle regulatory checkpoints and proliferate leading to cancer development and progression. Promoter DNA methylation contributes to carcinogenesis via transcriptional repression of tumor suppressor genes and/or transcriptional activation of oncogenes.

Reviewer 4 Report

Dear authors,

in the manuscript by Terisha Ghazi et al., you summarized the impact of dietary micronutrients, bioactive compounds, and food-borne mycotoxins on DNA methylation patterns and identified their potential in the onset and treatment of cancer.

I consider this work interesting and well organized. I think that this manuscript should be accepted to be published in this journal after little modifications. The authors should insert some figures with the chemical structures of some described compounds (for example resveratrol, genistein and so on).  

Author Response

Reviewer 4

 The authors should insert some figures with the chemical structures of some described compounds (for example resveratrol, genistein and so on).

The following Figures have been added to the manuscript:

Page 5 lines 123-127: “Accumulating evidence indicates that the human diet is a source of micronutrients (folate, B vitamins, betaine, choline, and methionine; Figure 2) and bioactive compounds (curcumin, epigallocatechin-3-gallate, genistein, quercetin, resveratrol, and sulforaphane; Figure 2) that act as both chemopreventative and chemotherapeutic agents by modulating the epigenome [13-16,18].

Page 6 line 131-133:

(PLEASE SEE FIGURE 2 IN UPLOADED REBUTTAL DOCUMENT)

Figure 2: Chemical structures of micronutrients and bioactive dietary compounds. Chemical structures were drawn using PubChem Sketcher Version 2.4.

Page 26 lines 644-645: “These mycotoxins include fusaric acid, fumonisin B1, deoxynivalenol, T-2 toxin, zearalenone, ochratoxin A, and aflatoxin B1 (Figure 4) [212].

Page 27 lines 650-652:

(PLEASE SEE FIGURE 4 IN UPLOADED REBUTTAL DOCUMENT)

Figure 4: Chemical structures of common food-borne mycotoxins. Chemical structures were drawn using PubChem Sketcher Version 2.4.

Round 2

Reviewer 1 Report

My comments concerned the text of Chapter 4 on mycotoxins.  

Line 630: The authors need to be more precise in there assertion. Constant exposure to HIGH LEVELS OF mycotoxins is common in areas where there are no regulations.Constant exposure to LOW LEVELS of mycotoxins cannot be avoided anywhere in the world.

Line 633-637: The authors need to emphasis that this paragraph is an introduction for the paragraphs after. Actually, the lecturer might be lost in the consecutive assertions of the actual three sentences that do not mention enough references and do not give sufficient precisions on the shown mechanism.

By the way, not all mycotoxins have an effect on the level of DNA methylation.

Consequently, I will suggest to delete the sentence line 633-634 and reformulate the following sentences. One way might be:

“Some of the main food-borne mycotoxins - fusaric acid, fumonisin B1, deoxynivalenol, T-2 toxin,

zearalenone, ochratoxin A, and aflatoxin B1-  have been shown to induce a change in DNA methylation profile in cell lines and model animals. The effects of these mycotoxins are presented in details here after….”

The Figure 5 – Is the scenario presented by the authors in the Figure 5 the mechanism that they are proposing or is this a mechanism already shown in which case the reference has to be cited? If the Figure 5 is a scenario presented by the authors, I will suggest to move it in the conclusion.

Author Response

The impact of natural dietary compounds and food-borne mycotoxins on DNA methylation and cancer

RESPONSE TO REVIEWERS

Reviewer 1

Line 630 (PDF): The authors need to be more precise in there assertion. Constant exposure to HIGH LEVELS OF mycotoxins is common in areas where there are no regulations. Constant exposure to LOW LEVELS of mycotoxins cannot be avoided anywhere in the world.

Page 26 lines 639-641: The following has been added “Mycotoxins are produced in response to several environmental factors such as warm humid conditions; and constant exposure to high levels of mycotoxins is common in areas where there are no regulations that protect the food intake of the populace [212].”

Lines 633-637 (PDF): The authors need to emphasis that this paragraph is an introduction for the paragraphs after. Actually, the lecturer might be lost in the consecutive assertions of the actual three sentences that do not mention enough references and do not give sufficient precisions on the shown mechanism. By the way, not all mycotoxins have an effect on the level of DNA methylation. Consequently, I will suggest to delete the sentence line 633-634 and reformulate the following sentences. One way might be: “Some of the main food-borne mycotoxins - fusaric acid, fumonisin B1, deoxynivalenol, T-2 toxin, zearalenone, ochratoxin A, and aflatoxin B1-  have been shown to induce a change in DNA methylation profile in cell lines and model animals. The effects of these mycotoxins are presented in details here after….”

Page 26 lines 643-644: The following sentence was deleted “The main food-borne mycotoxins are produced by Fusarium, Aspergillus, and Penicillium species.”

Page 26 lines 644-648: The following sentences “These mycotoxins include fusaric acid, fumonisin B1, deoxynivalenol, T-2 toxin, zearalenone, ochratoxin A, and aflatoxin B1 (Figure 4) [212]. Recent evidence suggests that mycotoxin-induced epigenetic modifications, in particular in the DNA methylation profile of the cell, can lead to neoplastic transformation and cancer development (Figure 5).” have been reformulated and now read “Some of the main food-borne mycotoxins – fusaric acid, fumonisin B1, deoxynivalenol, T-2 toxin, zearalenone, ochratoxin A, and aflatoxin B1 (Figure 4) – have been shown to induce a change in the DNA methylation profile in human cell lines and animal models [213-219]. The effects of these mycotoxins on DNA methylation are presented in detail hereafter.” – Page 26 lines 648-652.

Furthermore, additional references have been added to support the above information “Some of the main food-borne mycotoxins – fusaric acid, fumonisin B1, deoxynivalenol, T-2 toxin, zearalenone, ochratoxin A, and aflatoxin B1 (Figure 4) – have been shown to induce a change in the DNA methylation profile in human cell lines and animal models [213-219]. The effects of these mycotoxins on DNA methylation are presented in detail hereafter.” – Page 26 lines 648-652.

References

  1. Ghazi, T.; Nagiah, S.; Naidoo, P.; Chuturgoon, A.A. Fusaric acid-induced promoter methylation of DNA methyltransferases triggers DNA hypomethylation in human hepatocellular carcinoma (HepG2) cells. Epigenetics 2019, 14, 804-817, doi:10.1080/15592294.2019.1615358.
  2. Chuturgoon, A.; Phulukdaree, A.; Moodley, D. Fumonisin B1 induces global DNA hypomethylation in HepG2 cells – An alternative mechanism of action. Toxicology 2014, 315, 65-69, doi:https://doi.org/10.1016/j.tox.2013.11.004.
  3. Li, X.; Gao, J.; Huang, K.; Qi, X.; Dai, Q.; Mei, X.; Xu, W. Dynamic changes of global DNA methylation and hypermethylation of cell adhesion-related genes in rat kidneys in response to ochratoxin A. World Mycotoxin Journal 2015, 1, 1-12, doi:10.3920/WMJ2014.1795.
  4. Rieswijk, L.; Claessen, S.M.; Bekers, O.; van Herwijnen, M.; Theunissen, D.H.; Jennen, D.G.; de Kok, T.M.; Kleinjans, J.C.; van Breda, S.G. Aflatoxin B1 induces persistent epigenomic effects in primary human hepatocytes associated with hepatocellular carcinoma. Toxicology 2016, 350, 31-39.
  5. Karaman, E.F.; Ozden, S. Alterations in global DNA methylation and metabolism-related genes caused by zearalenone in MCF7 and MCF10F cells. Mycotoxin Res 2019, 35, 309-320, doi:10.1007/s12550-019-00358-8.
  6. Wang, H.; Zong, Q.; Wang, S.; Zhao, C.; Wu, S.; Bao, W. Genome-Wide DNA Methylome and Transcriptome Analysis of Porcine Intestinal Epithelial Cells upon Deoxynivalenol Exposure. Journal of Agricultural and Food Chemistry 2019, 67, 6423-6431, doi:10.1021/acs.jafc.9b00613.
  7. Liu, A.; Sun, Y.; Wang, X.; Ihsan, A.; Tao, Y.; Chen, D.; Peng, D.; Wu, Q.; Wang, X.; Yuan, Z. DNA methylation is involved in pro-inflammatory cytokines expression in T-2 toxin-induced liver injury. Food and Chemical Toxicology 2019, 132, 110661, doi:https://doi.org/10.1016/j.fct.2019.110661.

The Figure 5 – Is the scenario presented by the authors in the Figure 5 the mechanism that they are proposing or is this a mechanism already shown in which case the reference has to be cited? If the Figure 5 is a scenario presented by the authors, I will suggest to move it in the conclusion.

Figure 5 was prepared by the authors and depicts an overview of the proposed mechanism for how food-borne mycotoxins affect the DNA methylation profile of the cell thus leading to the development and progression of cancer. Figure 5 has been moved to the conclusion.

Pages 27-28 lines 656-663: The following was deleted

Figure 5: The effect of mycotoxins on DNA methylation and cancer. Mycotoxins can alter global DNA methylation and/or promoter DNA methylation in various cells both in vitro and in vivo. Global DNA hypomethylation leads to genome instability and increases the frequency of DNA mutations. Mutated cells are destroyed via apoptosis or evade cell cycle regulatory checkpoints and proliferate leading to cancer development and progression. Promoter DNA methylation contributes to carcinogenesis via transcriptional repression of tumor suppressor genes and/or transcriptional activation of oncogenes.”

Page 37 lines 944-946: The following was added “Figure 5 depicts an overview of the proposed mechanism for how food-borne mycotoxins affect the DNA methylation profile of the cell thus leading to the development and progression of cancer.”

Page 38 lines 951-958: The following was added

Figure 5: The effect of mycotoxins on DNA methylation and cancer. Mycotoxins can alter global DNA methylation and/or promoter DNA methylation in various cells both in vitro and in vivo. Global DNA hypomethylation leads to genome instability and increases the frequency of DNA mutations. Mutated cells are destroyed via apoptosis or evade cell cycle regulatory checkpoints and proliferate leading to cancer development and progression. Promoter DNA methylation contributes to carcinogenesis via transcriptional repression of tumor suppressor genes and/or transcriptional activation of oncogenes.”